# Mesozoic evolution of cicadas and their origins of vocalization and root feeding

Hui Jiang [1,2,3] ✉, Jacek Szwedo [4], Conrad C. Labandeira[5,6,7], Jun Chen [8], Maxwell S. Moulds [9], Bastian Mähler [3], A. Drew Muscente[10], De Zhuo[11], Thet Tin Nyunt [12], Haichun Zhang [1], Cong Wei [13], Jes Rust[3] & Bo Wang [1] ✉

Extant cicada (Hemiptera: Cicadoidea) includes widely distributed Cicadidae and relictual Tettigarctidae, with fossils ascribed to these two groups based on several distinct, minimally varying morphological differences that define their extant counterparts. However, directly assigning Mesozoic fossils to modern taxa may overlook the role of unique and transitional features provided by fossils in tracking their early evolutionary paths. Here, based on adult and nymphal fossils from mid-Cretaceous Kachin amber of Myanmar, we explore the phylogenetic relationships and morphological disparities of fossil and extant cicadoids. Our results suggest that Cicadidae and Tettigarctidae might have diverged at or by the Middle Jurassic, with morphological evolution possibly shaped by host plant changes. The discovery of tymbal structures and anatomical analysis of adult fossils indicate that mid-Cretaceous cicadas were silent as modern Tettigarctidae or could have produced faint tymbal-related sounds. The discovery of final-instar nymphal and exuviae cicadoid fossils with fossorial forelegs and piercing-sucking mouthparts indicates that they had most likely adopted a subterranean lifestyle by the mid-Cretaceous, occupying the ecological niche of underground feeding on root. Our study traces the morphological, behavioral, and ecological evolution of Cicadoidea from the Mesozoic, emphasizing their adaptive traits and interactions with their living environments.

Cicadas refer to the superfamily Cicadoidea, which are comprised of two modern families, Cicadidae and Tettigarctidae, whose monophyly is supported by molecular and morphological phylogenetic analyses of living species[1,2]. Extant Cicadidae consists of slightly more than 3000 worldwide species[2–7], while Tettigarctidae is thought to be a relict group containing only one living genus with two species in Australia[1,8–10]. Members of Cicadidae are well-known for producing some of the loudest sounds among insects through their tymbal

[1]State Key Laboratory of Paleobiology and Stratigraphy, Nanjing Institute of Geology and Palaeontology and Center for Excellence in Life and Paleoenvironment, Chinese Academy of Sciences, Nanjing 210008, China. [2]Institute of Geology and Paleontology, Charles University, Prague 12843, Czech Republic. [3]Section Palaeontology, Institute of Geosciences, Rheinische Friedrich-Wilhelms-Universität Bonn, Bonn 53115, Germany. [4]Laboratory of Evolutionary Entomology and Museum of Amber Inclusions, Department of Invertebrate Zoology and Parasitology, University of Gdańsk, Gdańsk PL80–308, Poland. [5]Department of Paleobiology, National Museum of Natural History, Smithsonian Institution, Washington, DC 20013, USA. [6]Department of Entomology and Behavior, Ecology, Evolution, and Systematics Program, University of Maryland, College Park, MD 20742, USA. [7]School of Life Sciences, Capital Normal University, Beijing 100048, China. [8]Institute of Geology and Paleontology, Linyi University, Linyi 276000, China. [9]Australian Museum Research Institute, Sydney, NSW 2010, Australia. [10]Princeton Consultants, Princeton, NJ 08540, USA. [11]Beijing Xiachong Amber Museum, Beijing 100083, China. [12]Department of Geological Survey and Mineral Exploration, Ministry of Natural Resources and Environmental Conservation, Myanmar Gems Museum, Nay Pyi Taw 15011, Myanmar. [13]Key Laboratory of Plant Protection Resources and Pest Management of the Ministry of Education, College of Plant Protection, Northwest A&F University, Yangling, Shaanxi 712100, China. ✉e-mail: huijiang2353@163.com; bowang@nigpas.ac.cn

mechanisms that consist of intricate organs producing sound through vibration of a ribbed membrane[11–13]. By contrast, Tettigarctidae is a clade that lacks production of loud sounds, and instead uses vibrational signals transmitted through the substrate for communication[14]. These two, considerably different, expressions of sound have led to speculation on the interpretation of tymbal structures associated with sound production and the evolution of their behaviours[8,14]. No previous work has explored the structural features and related behaviours associated with understanding the evolution of communications in cicada fossils.

Another remarkable feature of Cicadoidea is that their nymphs possess long-term and subterranean life habits, reflecting distinct ecological niches and survival strategies between their nymphal and adult stages. Cicada nymphs live underground, tunnelling through the substrate and feeding on xylem sap of roots, until the final instar emerges to undergo a final moult, subsequently emerging as a winged adult that achieves sexual maturity before its short-lived stage as an adult[15–20]. Cicada nymphs can live underground for up to 17 years[17,18],

with their life cycles producing significant effects on forest soils, microbial biomass, nutrient availability, predators, and host plants[21–25]. Immature and imaginal stages of individuals do not equally respond to the same evolutionary forces; therefore, different growth stages are of great significance in revealing different aspects of evolutionary mechanisms[26–28]. Consequently, nymphal fossils are necessary to illuminate the complete life cycles of ancient cicadas and their effects on terrestrial ecosystems, both below- and above ground. Nevertheless, cicadoid nymphal fossils are rare; only five incomplete and early instar nymphal fossils have been reported from mid- to Late Cretaceous and Cenozoic amber and an opal deposit[29–32]. These records clearly are insufficient to reflect the true evolutionary and ecological significance of cicadas as important soil and arboreal herbivores in insect history.

The adult fossil record of Cicadoidea includes 55 genera[33–40], which is largely represented by wing fragments, and consequently provides limited morphological information for early cicadoid lineages. Currently, all adult Mesozoic fossils are classified within Tettigarctidae, mainly based on forewing venation. Despite a few fossil sites

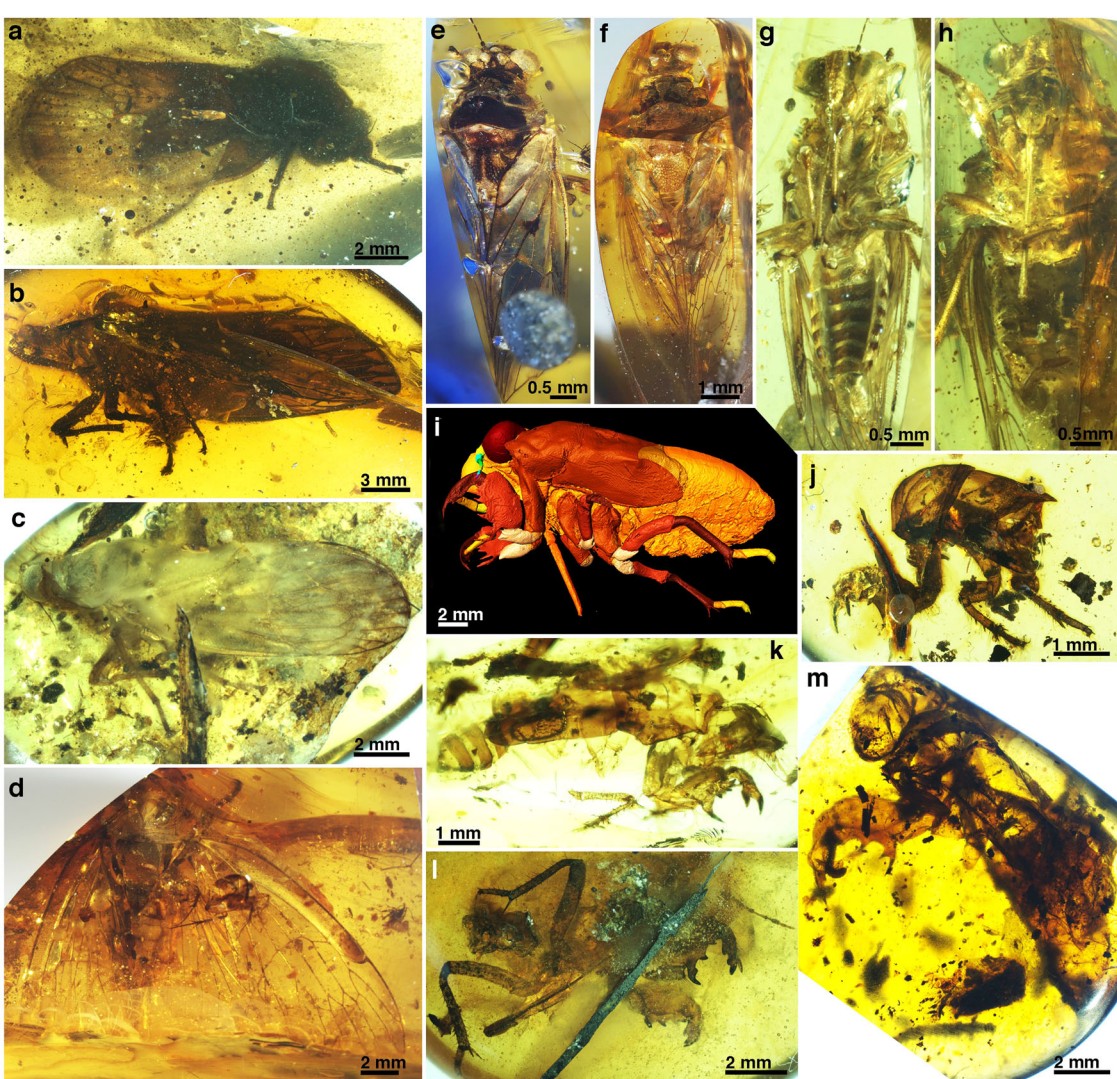

**Fig. 1 | Adults, final instar nymph, and exuviae of Cicadoidea fossils in Kachin amber of northern Myanmar. a** *Eunotalia emeryi* gen. et sp. nov. (MGM2016−014). This image was published in the study by ref. 41 (Fig. 3a). **b** *Cretotettigarcta problematica* comb. nov. (new material: NIGP201895). **c** *Cretotettigarcta shcherbakovi* sp. nov. (NIGP201896). **d** *Vetuprosbole parallelica* (new material: NIGP201897). **e**−**i** *Pranwanna xiai* gen. et sp. nov. (LYU−BC2001, male; LYU−BC2002, female). **e** Dorsal view of male. **f** Dorsal view of female. **g** Ventral view of male. **h** Ventral view of female. **i** Left view of final-instar nymph, Cicadoidea species 1 (NIGP2018985). **j**−**m** Final- nymphal exuviae. **j** Nymphal sp. 2 (MGM2016−017), left view. **k** Nymphal sp. 3 (LYU−BC2004), right view. **l** Nymphal sp. 4 (NIGP201900), ventral view. **m** Nymphal sp. 5 (NIGP201901), left view.

that preserve entire bodies of cicadoids, the present categorisation disregards many significant traits of these early fossils due to an over-reliance on traditional classification criteria and the lack of knowledge of continuous spatiotemporal morphological variation within this superfamily. To date, there is no clearly reviewed molecular evidence for assessing the divergence time between Cicadidae and Tettigarcti-dae. We think selecting the earliest currently known fossils of these two extant families for molecular clock calibration is disputable.

Here, we report newly described adults, a final-instar nymph, and exuviae of Cicadoidea from mid-Cretaceous (~99 Ma) Kachin amber that originated in northern Myanmar (Fig. 1). We use morphological data from fossil and extant Cicadoidea to conduct phylogenetic and morphological disparity analyses, trying to clarify the phylogenetic relationships between Mesozoic fossil and extant cicadoids and to illuminate macroevolutionary changes in their body structure adaptations. We discover the membranous tymbal and tymbal muscles associated with cicadoid sound production in the fossil record, and report fossil final-instar nymphs with specialised forelegs and long piercing-sucking mouthparts, indicative of both fossorial and root-feeding behaviours. In sum, we provide a more comprehensive picture of the relationships, and ecological and evolutionary history of early Cicadoidea.

## Results
### Systematic palaeontology
Order Hemiptera Linnaeus, 1758
 Suborder Cicadomorpha Evans, 1946
 Clade Clypeata Quadri, 1963
 Superfamily Cicadoidea Latreille, 1802

### Stem cicadoids
*Eunotalia* gen. nov.
 urn:lsid:zoobank.org:pub:8596E387-5FC6-4BA7-BBCA-B959D3BEC5E5

**Type species** *Eunotalia emeryi* sp. nov. by monotypy.

**Included species**: *Eunotalia emeryi* sp. nov.

**Etymology**: The generic name is a compound form, from Classical Greek prefix: *eu-*, meaning 'true' or 'good', and *notos*, meaning 'back' or 'dorsum'.

**Diagnosis**: Pronotum subhexagonal and enlarged, without a wrinkled collar, concealing all or part of mesonotum except scutellum, pronotal lateral angles pointed at about the middle length of pronotum. Paramedian fissure not distinct and lateral fissure absent, ambient fissure extending from both sides to the middle and almost intersecting the paramedian fissure sides, forming an acute angle. Forewing wide, costal area broad and with a fold. RA, RP, MA and MP forking into four, two, four and two branches, respectively. $CuA_2$ about half the length of $CuA_1$, following the nodal line but slightly separated and oriented backward, and terminating around nodal clave.

**Description:** as for the species.

**Remarks:** The new genus differs from other genera of Cicadoidea by unique combination of features described in the diagnosis.

*Eunotalia emeryi* gen. et sp. nov. (Figs. 1a, 2a, i and Supplementary Fig. 2).

 urn:lsid:zoobank.org:pub:8596E387-5FC6-4BA7-BBCA-B959D3BEC5E5

**Etymology**: The specific epithet is given in honour of Prof. David Emery from the University of Sydney, for his contribution to providing extant Tettigarctidae for anatomical study and his discussion of this research.

**Material**: Holotype: MGM2016−014, male. Silicified, description see in ref. 41. Deposited in the Myanmar Gems Museum, Nay Pyi Taw, Myanmar.

**Diagnosis:** As for the genus by monotypy.

**Description:** See Supplementary Note 1.1.

**Remarks:** Tymbal structure is identified in this fossil species (Fig. 2q). The abdomen is filled with mineral (Fig. 2y). Meracanthus has an enlarged base, and then tapering and narrowing forward; it is short and does not reach tergites II (Fig. 2ae). The thorax lacks identified opercula. The male genital includes identified structures such as anal style, median lobe of uncus and style (Fig. 2al).

### Stem cicadids
*Cretotettigarcta* ref. 34

 **Type species**. *Cretotettigarcta burmensis*[34].
 urn:lsid:zoobank.org:pub:8596E387-5FC6-4BA7-BBCA-B959D3BEC5E5

 *Hpanraais*[35], **syn. nov**.
 Type species *Hpanraais problematicus*[35].

**Included species**: *Cretotettigarcta bumensis*[34], *Cretotettigarcta problematica*[35] com. nov., and *Cretotettigarcta shcherbakovi* sp. nov.

**Material**: Holotype: NIGP167304, male, deposited in the Nanjing Institute of Geology and Palaeontology, Chinese Academy of Sciences; Nanjing, China.

**Diagnosis** (adapted from ref. 34): Pronotal lateral fissure present but weak; pronotum with wrinkled pronotal collar, almost the half length of the pronotum, lateral collar well-developed; pronotum concealed the part of mesonotum besides scutellum. The length of exposed mesonotum (except scutellum) no more than the half the length of pronotum; mesonotum with scutellum extending to around the end of abdomen. Apex of profemur with a robust lateral tooth-like spine; the upper medial part of protibia with a tooth-like spine; meso and metatarsi preserved two pairs of lateral spines, the second pair stronger than the first pair. Claw with fan-like arolium, arolium centre eminence with a two-petal ornamentation. Forewing with relatively short $RA_1$ and close to the nodal line; CuA with a strong arc bending shape at the base; M and CuA not fused or connected with short vein; $CuA_2$ sinuous and more than half the length of $CuA_1$.

**Description**: as in ref. 34.

**Remarks**: The mesoscutellum of *Cretotettigarcta* extending to the end of abdomen which is distinguished from other cicadoid genera.

*Cretotettigarcta problematica*[35] comb. nov. (Figs. 1b, 2c, k and Supplementary Fig. 3).

 urn:lsid:zoobank.org:pub:8596E387-5FC6-4BA7-BBCA-B959D3BEC5E5

**Material**: Holotype: NIGP168934, only one forewing. New material here: NIGP201895. Both are deposited in the Nanjing Institute of Geology and Palaeontology, Chinese Academy of Sciences; Nanjing, China.

**Diagnosis** (adapted from ref. 35): Male body length ~25.2 mm, width ~11.3 mm; the forewing length ~16.8 mm, width ~5.7 mm. Pronotal collar developed, spread out laterally, and lateral angle pointed. Separation of M closed to the separation of RP from R at the wing length. Crossvein *m-cu* connected $CuA_1$ and $MP_2$. Upper medial side of protibia with a relatively sharp spine.

**Description**: See Supplementary Note 1.2.

**Remarks**: New material of *C. problematicus* comb. nov. fits the revised diagnosis of *Cretotettigarcta*, with preserved body structures and long mesoscutellum.

*Cretotettigarcta shcherbakovi* sp. nov. (Figs. 1c, 2d, l and Supplementary Fig. 4).

 urn:lsid:zoobank.org:pub:8596E387-5FC6-4BA7-BBCA-B959D3BEC5E5

**Etymology**: The specific epithet is given in honour of Prof. Dmitry E. Shcherbakov from Paleontological Institute, Russian Academy of Sciences, for thanks to his contributions to the research of fossil Tettigarctidae.

**Material**: Holotype: NIGP201896, male, deposited in the Nanjing Institute of Geology and Palaeontology of the Chinese Academy of Sciences; Nanjing, China.

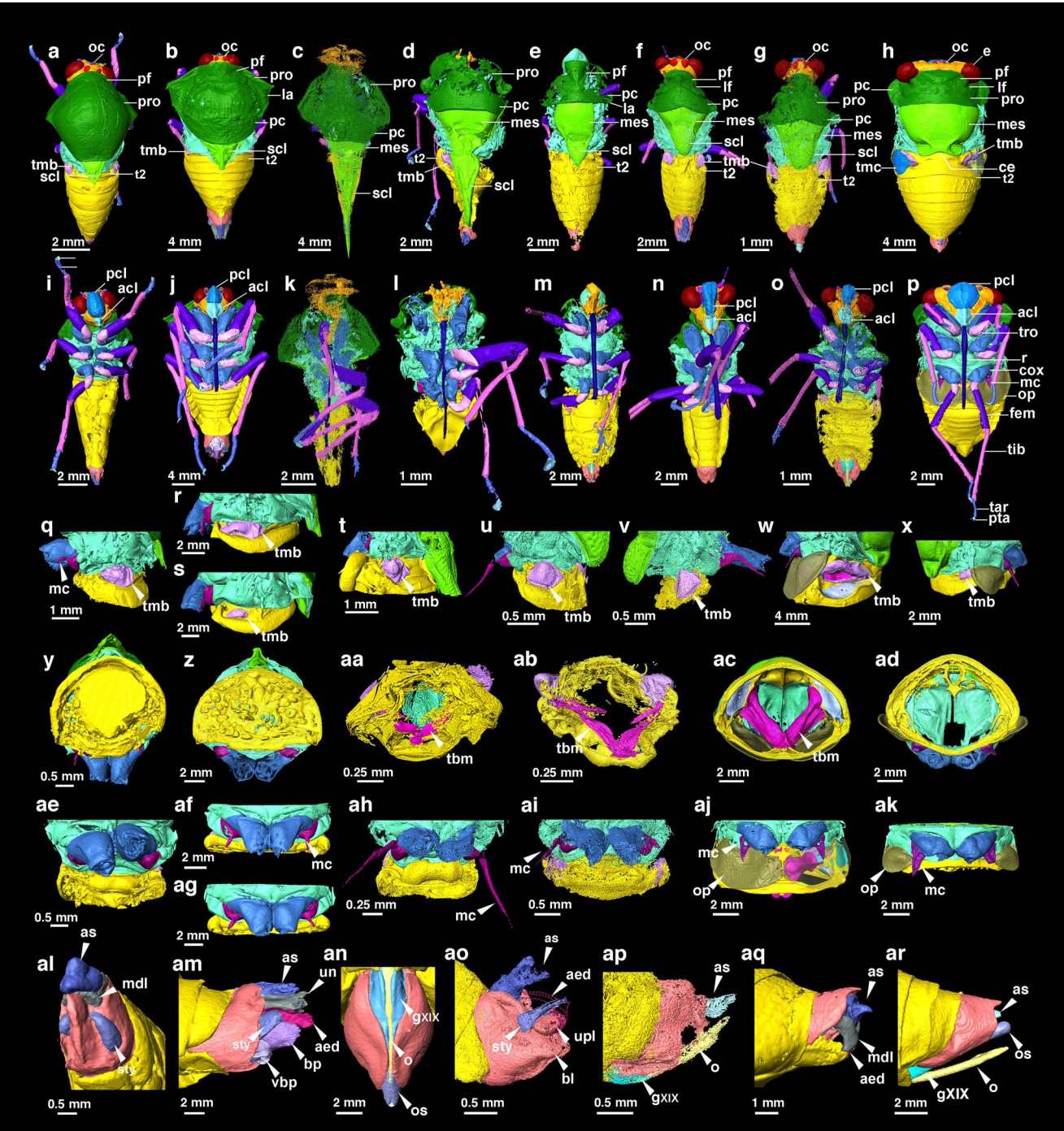

**Fig. 2 | Digital body structures reconstructed from micro-CT data showing the morphological evolutionary sequence of Cicadoidea. a–h** Dorsal view of *E. emeryi, Tettigarcta crinita* (extant Tettigarctidae)*, C. problematica, C. shcherbakovi, V. parallelica, Pr. xiai* and *Platypleura kaempferi* (extant Cicadidae). **i–p** Ventral view of (**a–h**). **q–x** Left view of partial thorax and first two segments of abdomen of *E. emeryi, T. crinita* (male)*, T. crinita* (female)*, C. shcherbakovi, Pr. xiai* (male)*, Pr. xiai* (female) and *Pl. kaempferi* (male)*, Pl. kaempferi* (female), showing tymbal structure in pink. For reflected light micrographs of extant *T. crinita* and *Pl. kaempferi* with associated tymbal organs (see Supplementary Fig. 18). **y–aa** Rear view of transverse section of the second segment of the abdomen of *E. emeryi, T. crinita* (male), and *Pr. xiai* (male), showing the interior condition of the abdomen. **ab** Frontal view of a transverse section of the first segment of the abdomen, showing the interior of the abdomen, vertical view of *Pr. xiai* (male). **ac–ad** Rear view of transverse section of second segment of the abdomen of *Pl. kaempferi* (male)*, Pl. kaempferi* (female), showing the interior of the abdomen. **ae–ak** Ventral view of a partial thorax and two segments of the abdomen in (**q–s, u–x**). **al–ar** Genitalia of (**q–s, u–x**). The structure 'aed' in ao represented by the lines of dashes is drawn based on observations of the specimen under the microscope (Supplementary Fig. 1t, u). acl anteclypeus, aed aedeagus, as anal style, bl basal lobe of pygofer, bp basal plate, ce cruciform elevation, cox coxa, e compound eye, fem femur, gXIX gonocoxite XIX, lf lateral fissure, mc meracanthus, mdl median lobe of uncus, mes mesonotum, o ovipositor, op operculum, os ovipositor sheath, pc pronotal collar, pcl postclypeus, pf paramedian fissure, pro pronotum, pta ptarsus, r rostrum, sty style, t2 tergite 2, tar tarsus, tbm tymbal muscle, tmb tymbal, tmc tymbal cover, tib tibia, tro trochanter, scl scutellum, un uncus, upl upper lobe of pygofer, and vbp ventrobasal pocket. Each colour in the image represents a different structure and these colour-structure associations remain consistent throughout the figure even when the label is not repeated in each panel.

**Diagnosis**: Male body length ~13.6 mm, width ~5.9 mm; the forewing length ~11.1 mm, width ~3.9 mm. Distance between the compound eyes approaching the 1/2 diameter of the compound eyes. Pronotal lateral angle blunt, concave in the middle of pronotal posterior margin not distinct. Crossvein *m-cu* connected between $CuA_1$ and forking point of MP. Upper medial side of protibia with a relatively blunt spine.

**Description**: See Supplementary Note 1.3.

**Remarks**: Crossvein *m-cu* connects to $CuA_1$ and forking point of MP is distinguished in the new species from the other known species in this genus. The tymbal structure is identified in this fossil species (Fig. 2t). The meracanthus is not preserved completely, but it exhibits an enlarged base (Fig. 2l). The abdomen is deformed.

### *Vetuprosbole*[34]
**Type species**: *Vetuprosbole parallelica*[34], by monotypy.
urn:lsid:zoobank.org:pub:8596E387-5FC6-4BA7-BBCA-B959D3BEC5E5

**Included species**: *Vetuprosbole parallelica*[34].

**Material**: Holotype: NIGP168021, gender unknown. New material here: NIGP201897, female (Figs. 1d, 2e, m and Supplementary Fig. 5). Both are deposited in the Nanjing Institute of Geology and Palaeontology, Chinese Academy of Sciences; Nanjing, China.

**Diagnosis** (adapted from ref. 34): Pronotum concealing small part of mesonotum besides scutellum, with wrinkled collar. Pronotal lateral fissure relatively weak. Length of exposed part of mesonotum (except scutellum) slightly shorter than the length of pronotum. Mesonotum with scutellum extending near abdominal tergite II. Metatarsus preserved two pairs of lateral spines, and second pair of spines more developed than the first pair. Claw developed with fan-like arolium. Forewing RA forking into three branches and $RA_1$ next to Sc and split before the margin. $CuA_2$ long and sinuous, more than 2/3 length of $CuA_1$. CuP fused a short segment with 1A at the basal part.

**Description**: as for the species of a new specimen. See Supplementary Note 1.4.

**Remarks**: New features found in the new material of *V. parallelica* include a wrinkled collar that is less than half the length of the pronotum, a larger exposed mesonotum with a scutellum, and CuP fused to a segment with 1A. These features allow it to be distinguished from all other known genera of Cicadoidea. The meracanthus of *V. parallelica* is triangular and reaches tergites II (Supplementary Fig. 5c). The thorax lacks identified opercula. The female genital includes identified structures such as gonocoxite XIX, ovipositor and ovipositor sheath.

### *Pranwanna* gen. nov
urn:lsid:zoobank.org:pub:8596E387-5FC6-4BA7-BBCA-B959D3BEC5E5

**Type species** *Pranwanna xiai* sp. nov., by monotypy.
**Included species**: *Pranwanna xiai* sp. nov.
**Etymology**: The generic name, *pranwanna*, is from the Jingpho language spoken in Kachin State of Myanmar, meaning 'primitive'.

**Diagnosis**: Pronotum sub-trapezoidal; front edge forms an open, neck-like feature; the middle of posterior margin concave, resembling W-shape of the entire posterior margin; wrinkled collar half-length of pronotum. Paramedian and lateral fissures well-developed. Pronotum concealing part of mesonotum. Length of exposed mesonotum slightly shorter than the length of pronotal collar; mesonotum with inflated scutellum. rounded distally, extending near abdominal tergite, scutoscutellar sulcus and base of lateral ridge prominent; two dark brown bands on the surface of scutellum in male specimen, and the bands widened gradually from the centre to both sides; female not observed the band pattern on scutellum. No developed and dilated opercula. Forewing with separation of M and the separation of RP from Sc+R around the nodal; MA forking into two branches and MP single; common stem of M+CuA long; $CuA_2$ short and straight. 1A fused a short segment with CuP. 2A absent. Nodal line relatively weak. Both

male and female specimen possessing tymbal; male with robust tymbal muscles and developed abdominal cavity.

**Description:** as for the species.

**Remarks:** *Pranwanna* gen. nov. is the smallest cicadoid in the fossil record so far. The distinct lateral fissure; scutellum distal aspect rounded and blunt, 2A absent, with long common stem of M+CuA distinguish the new taxon from other known genera of Mesozoic cicadoids.

***Pranwanna xiai* sp. nov.** (Figs. 1e–h, 2e, f, n, o and Supplementary Figs. 6, 7)

urn:lsid:zoobank.org:pub:8596E387-5FC6-4BA7-BBCA-B959D3BEC5E5

**Etymology**: The specific epithet is given in honour of Mr. Fangyuan Xia from the Lingpoge Amber Museum for his contribution to discovery the holotype and allotype specimens.

**Material**: Holotype: LYU–BC2001, male. Allotype: LYU–BC2002, female. Paratype: LYU–BC2003, male. Type specimens are deposited in the Linyi University, Linyi, China.

**Diagnosis**: As for the genus by monotypy.

**Description**: See Supplementary Note 1.5.

**Remarks:** Tymbal and tymbal muscles are identified in this fossil species (Fig. 2u, v, aa, ab). The abdomen has a cavity and preserves malpighian tubules. (Supplementary Fig. 7). Meracanthus is quite long in male, beyond half the length of tergites II of abdomen; meracanthus in female reaches approximately tergite II of abdomen (Fig. 2ah, ai). The thorax lacks identified opercula (Fig. 2ah, ai). The male genital includes identified structures such as anal style, aedeagus, upper lobe of pygofer, basal lobe of pygofer and style (Fig. 2ao). The female genital includes identified structures such as anal style, gonocoxite XIX and ovipositor (Fig. 2ap).

## Phylogeny and morphological disparity

Here, we perform morphological phylogenetic analyses of the Cicadoidea, integrating both fossils and extant taxa. The data matrix was analysed using the maximum parsimony method and Bayesian inference. We synthesised morphological information from both published extant and fossil species and supplemented this with micro-CT scan anatomical data from six newly reported adult amber fossils and four extant specimens included in this study. This examination resulted in a total of 81 morphological characters that were used to reconstruct the phylogenetic trees (Supplementary Note 2, Supplementary Data 1 and Supplementary Fig. 8). In this study, we primarily aim to establish phylogenetic relationships between the extant cicadoids and fossil groups discovered in the Middle Jurassic Daohugou deposit, Inner Mongolia, China, and in mid-Cretaceous Kachin amber in Northern Myanmar. The reason behind our selection of fossils from these two localities for phylogenetic reconstruction is attributed to the discovery of a relatively rich and morphologically complete collection of Cicadoidea fossils. These fossils, preserving both winged and non-winged body structures, enable us to establish a precise morphological correspondence between the winged and non-winged parts within the same species.

The main results from maximum parsimony and Bayesian inference are relatively congruent (Supplementary Fig. 8). In both analyses, *Eunotalia* gen. nov. was recovered as the stem group of the cicadoids. *Tianyuprosbole zhengi* was recovered as the sister-group of the extant tettigarctids. *Shuraboprosbole*, *Sanmai*, and *Macrotettigarcta* from Daohugou, as well as *Cretotettigarcta*, *Vetuprosbole*, and *Pranwanna* gen. nov. from Kachin amber were recovered into a large clade with extant cicadids. In the results of maximum parsimony, *Shuraboprosbole*, *Sanmai*, and *Macrotettigarcta* cluster as one clade, although this is not strongly supported. In the results of Bayesian inference, these genera are shown to constitute a paraphyletic group. The groups, *Cretotettigarcta*, *Vetuprosbole*, and *Pranwanna* gen. nov., were recovered closer to the extant cicadids. *Pranwanna* gen. nov. was

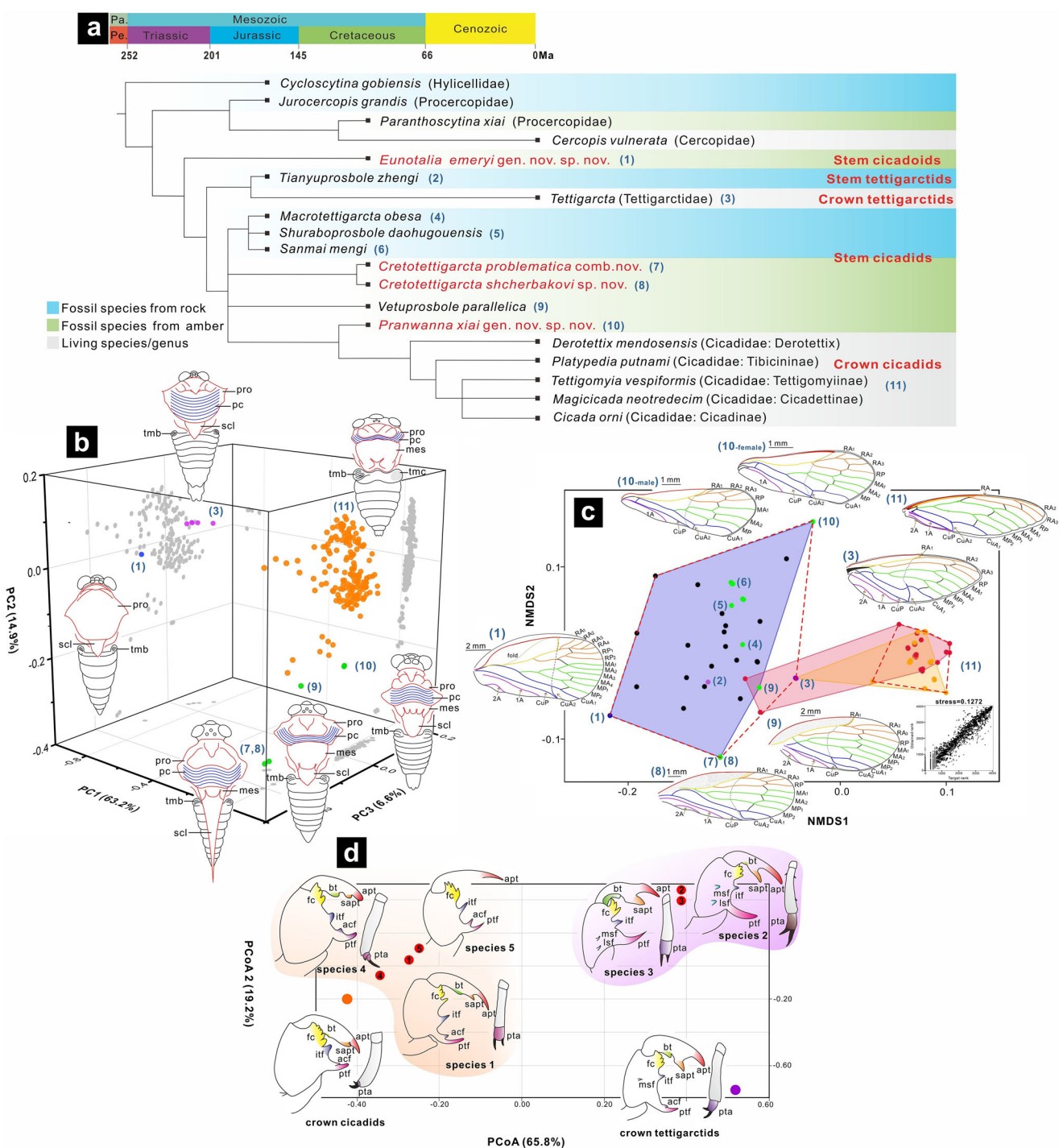

**Fig. 3 | Lineage reconstruction based on data from a phylogenetic analysis and morphological diversity. a** Schematic diagram of the phylogenetic relationships from the analysis of TNT (Bootstrap values >50%), showing phylogenetic positions of cicadoid fossils. Names in red indicate newly described species herein or a new taxonomic placement. Complete phylogenetic trees, see Supplementary Fig. 8. **b** 3D ordination plot of geometric morphometrics (GM) analysis of the dorsal profile of the head and notum of Cicadoidea. Blue area includes the body macro-morphological data of the Mesozoic cicadoid fossils, and pink area includes the body macromorphological data of the extant cicadoid species. Line drawings referenced from refs. 3,9. **c** Ordination plot of non-metric multidimensional scaling (NMDS) of an analysis of cicadoid forewings. The numerical labels of the line drawings and points in (**b**) and (**c**) correspond to species names numbered in (**a**). Blue area, yellow area, and pink area includes forewing macromorphological data of the Mesozoic cicadoid fossils, Cenozoic fossils and extant cicadoid species.

**d** Ordination plot of principal coordinates analysis (PCoA) of nymphal legs, showing forelegs and foretarsi of Cicadidae, Tettigarctidae, sp. 1, sp.3 and sp.4, foreleg and metatarsi of sp. 2, and only foreleg of sp. 5. For characteristic descriptions, matrixes and additional explanatory data, see Supplementary Materials. Abbreviations: acf accessory tooth of femur, apt apical tooth of tibia, bt blade of tibia, clw pretarsal claw, cox coxa, fc femoral comb, fem femur, itf intermediate tooth of femur, lsf lower lateral spine on the outer surface of femur, msf mid-lateral spine on the outer surface of femur, pc pronotal collar, Pa. Palaeozoic, Pe. Permian, pta pretarsi, pro pronotum, ptf posterior tooth of femur, sapt secondary apical tooth of tibia, scl scutellum, tar tarsus, tib tibia, tro trochanter, tmb tymbal and tmc tymbal cover. Each colour in the image represents a different structure, and these colour-structure associations remain consistent throughout the figure even when the label is not repeated in each panel.

recovered nearest to the clade of extant cicadids. The clearly relationship between *Cretotettigarcta* and *Vetuprosbole* was not strongly supported in both analyses. Considering that any evolutionary clade can be divided into a stem group and a crown group in theory[42], and to better illustrate the relationships between the fossils and extant taxa, we provide a simplified schematic of the phylogenetic results in Fig. 3a.

To facilitate the identification of small changes in morphological structures, we also quantified specific body morphology using continuous or categorical variable features data, separately considering aspects such as non-winged adult body structures, wing venation and outline features, and key nymphal leg characters. Different data analysis protocols, including a principal component analysis (PCA), nonmetric multidimensional scaling (NMDS), and a principal coordinate analysis (PcoA), have been utilised to perform morphological disparity analysis of the body structures[43–49].

Through morphological observations, we have discovered that some homologous non-wing body features present different results across various species. However, these characters have often been overlooked in previous identifications, such as the distance between the compound eyes, the length of the pronotum, the collar of the pronotum, the position of the anterior lateral angles of the pronotum relative to pronotum length, the exposed length of the mesonotum and the terminal morphology of the mesonotum. We selected ten landmarks to quantify the relative positions of these structures (Supplementary Fig. 9 and Supplementary Table 1). Figure 3b shows the PCA results of the morphospace for the dorsal features of the head and thorax, obtained through geometric morphometrics. The first three principal components account for 63.2%, 14.9% and 6.6% of the variation, with the cumulative variation explaining 84.7% of the total shape variance of dorsal features of the head and thorax, whose support these features could reflect the main pattern of variation in differences of the dorsal profile of the head and thorax (Fig. 3b, Supplementary Tables 2 and 3).

Because forewing morphology plays a crucial role in the taxonomic study of insect fossils, analysing the morphospace of the forewing assists in visualisation of morphological similarities, facilitating the analysis of morphological differences and comparisons with phylogenetic outcomes. Figure 3c illustrates the result of morphological disparity obtained by quantifying the forewing features within the Cicadoidea. To ensure an unweighted and comprehensive selection of features, we chose 32 characters, including as many forewing classificational features as possible (Supplementary Note 3 and Supplementary Data 2). We employed a method that describes discrete data to quantify forewing characters and utilised NMDS for data analysis. The stress value ie 0.1272 indicates that the result has a good reliability level. The similarity among forewings can be judged by the distance among the scatter points to some extent (Fig. 3c). The forewings of Mesozoic cicadoid fossils exhibit a higher similarity to extant crown tettigarctids. The overall similarity of the forewings of modern tettigarctids is closer to those of crown cicadids than most Mesozoic cicadoids are to crown cicadids. *Pranwanna* gen. nov shows a greater resemblance to the forewings of the crown cicadids than other known Mesozoic groups. The morphospace of forewing morphology in the Mesozoic is greater than that in the Cenozoic, which is, in turn, greater than that of modern groups. Mesozoic groups, particularly the stem groups of the Cicadoidea, are located on the far left of Fig. 3c. Following that, the stem cicadids appears, with the crown cicadids situated on the far right of Fig. 3c. The distribution of morphological similarity is consistent with phylogenetic relationships and temporal evolution, indicating that the evolution of wing morphology has a certain directionality.

From the Mesozoic fossil groups to modern crown groups, a morphological evolutionary trend in wing profile also seems present. Therefore, we selected nine landmarks to calibrate the wing profile individually and conducted a PCA (Supplementary Figs. 10a, 11 and Supplementary Table 4). The results support the major variation pattern of forewing profile difference to a certain extent (Supplementary Fig. 10a and Supplementary Tables 5, 6). The morphological changes in forewing profiles from the Mesozoic groups to modern crown groups are mainly reflected in changes in the curvature of the wing profile, which overall is transformed from a rectangular ellipse to a triangle (Supplementary Fig. 11).

Moreover, to further assist in understanding the morphological changes, we attempted to gather and compare statistics on body length and width, forewing length and width, length of the prothoracic and exposed mesothoracic notum, distance between the two compound eyes, width of head, width of the base of the labium, and width of the rostrum from both fossil and extant groups. The statistical trend shows that,wing length and width, body length and wing length and body length and width had proportional relationships (Supplementary Fig. 10b, c). From the analysis of measurement data from both fossil and modern species, we found that overall body length does not show a pattern of increase from the Mesozoic to the modern cicadoids. The results indicate that the body length and wing length of both fossil cicadas and extant cicadas fall within a relatively stable range, specifically between 5.45–64.6 mm and 5.6–78.5 mm (Supplementary Fig. 10b, c and Source Data), respectively, although the body width of many compressed fossils cannot be accurately measured due to preservation issues. However, both statistical analysis and morphological observations indicate that crown group cicadas generally display a tendency towards wider bodies, evident in features such as increased head width, greater distances between compound eyes (Supplementary Fig. 10d and Figs. 2a–h, 3b), wider clypeus (Supplementary Fig. 10e and Fig. 2i–p), and the appearance of a central groove in the postclypeus (Fig. 2p). From the discovery of stem groups to modern crown groups of Cicadoidea, there is a noticeable overall macroevolutionary trend of increased exposure of the mesonotum and a shortening of the pronotum (Supplementary Fig. 10f and Figs. 2a–h, 3b).

## Morphological analysis of nymphal fossils

The five nymphal specimens (Fig. 1i–m) are certainly placed in Cicadoidea, based on their well-developed capsular compound eyes, prominent postclypeus, long piercing-sucking mouthparts, and specialised fossorial forelegs with developed femoral combs. The distinctive features of each specimen are sufficient to classify these individuals into five species (Supplementary Note 4 for descriptions and Supplementary Data 3 for a comparative summary). However, it is a challenge to clearly connect fossil adults to nymphs in the separated amber pieces. To facilitate the correspondence of these nymphal fossils with adult fossils in a future study, we provisionally categorise them as undetermined Cicadoidea species 1 (NIGP201898, Fig. 1i), species 2 (MGM2016-0.17, Fig. 1j), species 3 (LYU-BC2004, Fig. 1k), species 4 (NIGP201900, Fig. 1l), and species 5 (NIGP201901, Fig. 1m). Species 1 is a final-instar nymph (Supplementary Fig. 12) and spp. 2 to 5 are exuviae retaining features of the final instar nymphs (Supplementary Figs. 13–15).

The legs have the most distinctive identifying features of these fossil nymphs (Supplementary Figs. 12, 14, 15, 16). Species 1, 4 and 5 share several characteristics with extant Cicadidae, including the profemur with an accessory tooth, and the secondary apical tooth of tibia that are small, obtuse, and no longer than 1/2 length of apical tibial tooth (Supplementary Fig. 14a–i). Species 2 and 3 show the typical features of extant Tettigarctidae, including the secondary apical tooth of tibia of the profemur are much longer and more robust, resembling a giant pair of pliers, and no less than 1/2 the length of apical tibial tooth separated from the apical tooth by a certain distance (Supplementary Fig. 14k–p). The unique features of species 2 and 3 namely, include two lateral spines on the outer surface of the profemur, two rows of spines on both sides of the mesotibia, a dominant and prong-like spine among apical spines on the mesotibia and metatibia, and a

metatarsus with lateral spines. The examined fossils present four or five teeth of the femoral comb. Final-instar nymphs of extant Tettigarctidae display three teeth of the femoral comb whereas the final-instar nymph of extant Cicadidae display three to nine teeth of the femoral comb[50] (Supplementary Data 3). Most species of Cicadidae display six to nine comb teeth, which seems to be a derived character of Cicadidae. The terminal teeth of the femoral comb in extant cicadoids obviously are aggregated into a large flat plate; while this feature is not evident in the current fossils, it seems to indicate distinctive digging abilities. Additionally, other features including the reach of the mouthparts, the diameter and shape of last antennal article, and development of the point of the tibial blade, as detailed in Supplementary Data 3, also are noteworthy. These nymphal traits varied among fossil and living species, which could be useful for future taxonomic and phylogenetic analyses.

We conducted a morphological disparity analysis, a principal coordinates analysis (PCoA), on the leg features of cicada nymphs (Fig. 3d). To prevent discrepancies resulting from structural deformations in the captured images, we quantified the leg features using descriptive discrete data (Supplementary Table 7). PCoA results display the morphological disparity between nymphs from mid-Cretaceous taxa to extant taxa. These results show that the morphological similarities of the nymph species 1, 4 and 5 are closer to crown cicadids, and that the morphological similarities of nymphal species 2 and 3 are closer to crown tettigarctids when compared with crown cicadids (Fig. 3d and Supplementary Table 8).

We have also compared the five nymphs that have been found in the fossil record so far (Supplementary Fig. 16). Because both living Cicadidae and Tettigarctidae possess an accessory femur tooth (Fig. 3d and Supplementary Fig. 14q–t), it's not a reliable characteristic to classify fossils into these two families. As a result, using this trait for classification makes it unreliable to place the earlier instar nymphs found in Kachin amber[31], late Eocene Baltic amber[29], and mid-Miocene Dominican amber[31] within the crown cicadid clade. The micro-CT data of the early Pliocene nymph from Indonesia shows that the intermediate and first teeth are developed on the femoral comb[32], indicating that this specimen does not represent a first instar nymph[51]. The nymphal leg fragments found in Late Cretaceous New Jersey amber, shown in a line sketch[30], exhibit a developed femoral comb that displays four teeth, indicating that they belong to a final-instar nymph and further confirming the generally low number of comb teeth in early cicada nymphs. Other features of this specimen[12], such as two confirmed stout spines on the outer face of the femur and a large secondary apical tooth on the tibia that is separated from the tibial apical tooth by a certain distance, show similarities with spp. 2 and 3. Enlarged pretarsus is found in the nymphs of Mosozoic fossil spp. 1–3 and extant tettigarctids[34,52], but not yet in extant cicadids and nymphal sp. 4 fossil. Also associated with adults, an arolium and empodium are present in Mesozoic adult fossils and extant Tettigarctidae respectively, while extant Cicadidae lacks these structures[1,8]. We infer that the enlarged pretarsus in the nymphs is manifested as an arolium in the adults, which is a plesiomorphic feature of Cicadoidea. The empodium is an apomorphy of extant Tettigarctidae. Overall, most of the nymphal features of extant Cicadidae and Tettigarctidae have existed since the mid-Cretaceous.

## Discussion
### Phylogenetic relationships, estimation of divergence time and morphological macroevolution patterns
The phylogenetic results demonstrate that before 'Tettigarctidae' which includes Mesozoic fossils formed a paraphyletic group (Fig. 3a). Some Mesozoic cicadoid fossils[34,53,54], including *Macrotettigarcta*, *Shuraboprosbole*, and *Sanmai*, *Cretotettigarcta*, *Vetuprosbole* previously classified as Tettigarctidae may, in fact, belong to stem cicadids. *Tianyuprosbole* might be a group within stem tettigarctids[55]. The

clade of modern tettigarctids diverged from the clade of modern cicadids prior to the divergence of many Mesozoic fossils from the clade of modern cicadids. Stem groups of cicadids and tettigarctids found in the Middle Jurassic Daohugou beds of Inner Mongolia, China, indicate that the ancestral lineages of Cicadidae and Tettigarctidae diverged by at least the Middle Jurassic. *Eunotalia emeryi* gen. et sp. nov., identified as a stem cicadoid in Kachin amber, suggests the possibility of inferring characteristics of more basal groups within the Cicadoidea from Cretaceous fossils.

The results combined from Fig. 3a, b suggest that *Eunotalia emeryi* gen. et sp. nov. is a stem cicadoid, and exhibits a feature similar to that of stem and crown tettigarctids, in that its pronotum entirely covers the mesonotum except for the small scutellum. In contrast, in both stem and crown cicadids, the pronotum is shortened and exposed mesonotum is enlarged. This suggests that the larger pronotum covering part of the mesonotum is primitive and represents a plesiomorphic trait of Cicadoidea. Other features, such as the absence of a wrinkly collar, the absence of the lateral fissure, and presence of the fold in the forewing costal area are identified in *Eunotalia emeryi* gen. et sp. nov. These features may represent primitive characteristics of early cicadoids. Stem cicadids share some traits with crown cicadids, such as developed paramedian and lateral fissures, an exposed and larger mesonotum (minus the scutellum), and a wrinkled collar whose length is no longer than one-half that of the pronotum. These morphological features also can be somewhat observed in Jurassic compression fossils (Supplementary Fig. 17), although these features have not previously received much attention or been considered significant. The similarity of dorsal view features of the head and thorax of *Pranwanna* gen. nov. and *Vetuprosbole* are closer to the crown cicadids compared to *Cretotettigarcta* (Fig. 3b). The most distinctive, influencing feature is the distal part of the mesonotum. This structure in crown cicadids show a cross-like shape in dorsal view generally called the cruciform elevation, whereas the distal part of mesonotum of the rest of cicadoid fossils exhibit predominantly an equilateral triangle shape in dorsal view, with (almost) no splitting at the distal end, and is generally called a scutellum. The cruciform elevation and scutellum are homologous structures[1], but with different morphological profiles. In the vast majority Cicadoidae species, this structure is generally reached in the first two segments of the abdomen, however the mesoscutellum of *Cretotettigarcta* can extend to the end of the genitalia which had never been recognised in previously in the fossil record and never has been found in modern groups of Cicadoidae. Taking into account the morphological observation and statistical analysis results (Fig. 2 and Supplementary Fig. 10), we hypothesise that the cruciform elevation may be derived from the distal separation of the mesoscutellum, a consequence of the body widening. Usually, the wings of cicadas are fixed on both sides of the scutellum (or cruciform elevation) when resting, and the changes in this structure may be associated with the change in wing morphology and flight movement ability.

The results of forewing morphological disparity and forewing profile geometric morphometrics also clearly indicate a significant evolutionary trend in the forewing morphology of the Cicadoidea from Mesozoic fossil records to extant crown groups (Fig. 3c). The wings of Mesozoic fossils are more similar to those of extant tettigarctids than to those of extant cicadids. The large dashed-line enclosure in Fig. 3c shows the potential classification of species within 'Tettigarctidae' including fossils, based on the original standards derived from analyses of certain forewing characters. The small dashed-line enclosure shows the classification for the previously identified 'Cicadidae' including Cenozoic cicadids. This explains why previous studies, relying solely on forewing characteristics, easily assigned Mesozoic fossils to extant tettigarctids. These assignments supported the view that extant tettigarctids retain the plesiomorphic characteristics of Mesozoic fossils, such as a distinct costal area, separation of CuP and 1A, the length of $CuA_2$ not less than half the length of $CuA_1$, a distinct

nodal line, RP separation from R earlier than the forking point of M, and the angle of the anal area of the wing lower than the cubital angle. However, other features have a clear temporal trend from the basal group of Cicadoidea to the extant crown group, such as a costal area narrowing, a shortening and narrowing of the $RA_1$ vein, and a $CuA_2$ vein not extending towards the wing apex. The afore-mentioned features also is why the overall forewing similarity of extant crown tettigarctids is closer to the cicada crown group than most Mesozoic groups, as shown in Fig. 3c. The overall wing shape also shows a morphological evolutionary trend from a rectangular ellipse to a triangle form from the Mesozoic to the modern, cicadid-dominated crown group (Supplementary Figs. 10a, 11). Nonetheless, both forewing venations and overall wing shape indicate a distinct change in flight ability[56,57]. Combining the results from Fig. 3b, c, it is evident that the changes in the dorsal thoracic notum are closely related to the change of the forewing venation and profile. From the basal groups to the crown groups, the overall body form shows a trend of reduction of the pronotum and enlargement of the exposed mesonotum. As the forewings are connected to the mesothorax, the enlargement of the exposed mesonotum may suggest enhancement of the thoracic flight muscles, which appears also to indicate a transformation in wing morphology and flight capabilities. This trend also reveals the obvious difference in the appearance of the thoracic notum of extant tettigarctids and cicadids from an evolutionary perspective. The genus *Pranwanna* gen. nov. exhibits a developmental trend in body and forewing morphology closer to crown cicadids compared to other Mesozoic cicadoids, suggesting that the transformation in wing flight capabilities of cicadoids may have been present during the mid-Cretaceous. Our research demonstrates that highly specialised, homologous body structures in insect fossils may contain identifiable transitional variants previously have been overlooked. Meticulous investigation of these continuous morphological transformations may allow for a more precise understanding of the influence of temporal and spatial changes on morphological evolution and further assist in elucidating the patterns of macroevolution.

Fossil nymph features were not included in the phylogenetic analysis. Considering the low number of discovered nymph fossils, along with their preservational quality, their characters are insufficient for precise correlation with the adult fossils found in Kachin amber. However, the leg morphology comparisons and the results from Fig. 3d indicate that leg features of cicadoids bear significant taxonomic implications, supporting the phylogenetic result that the crown tettigarctids and cicadids differentiated before the mid-Cretaceous interval. Nymphal species 1, 4 and 5 display a morphological similarity that are closer to the crown cicadids than to crown tettigarctids, suggesting they may belong to the stem cicadid groups. Considering other characteristics of the nymphal body and the classification position of the adult fossils from Kachin amber, Nymphal species 2 and 3 exhibit a high morphological similarity, implying a close phylogenetic relationship and they resemble crown tettigarctids more than the crown cicadids. In comparison to the results obtained from adult fossils, our study suggests that isolated research on nymph fossils can also provide relatively reliable morphological evidence for taxonomy and the times of clade separation.

Our results indicate that Cicadoidea had a greater spectrum of morphology during the Mesozoic, revealing a more complex and diversified evolutionary pattern for Mesozoic Cicadoidea (Fig. 4). The phylogenetic results suggest that Mesozoic fossils encompass many stem groups of extant lineages. Relying solely on qualitative descriptions of wing or body-part morphology for taxonomic features may not fully clarify the phylogenetic relationships between fossil insects and extant groups. These approaches highlight potential issues with the strategic classification that directly assigns Mesozoic insect fossils into extant groups. Such a practice could lead to an inadequate understanding of the relationships between fossil records and extant

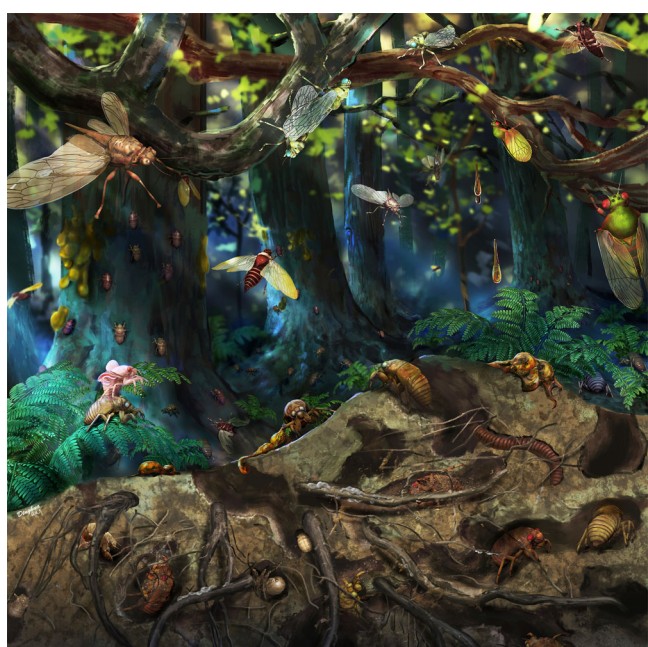

**Fig. 4 | Life reconstruction of cicadas in a Mesozoic Forest.** Reconstructed by Mr. Dinghua Yang. For individual reconstructions of the fossils see Supplementary Fig. 20.

taxa that potentially overlook many transitional features in evolution, making it difficult to present a more continuous evolutionary trajectory. Nevertheless, our study initially has established a phylogenetic framework for Cicadoidea based on the adult forewing and non-wing body structures, as well as comparisons of body structure morphospace features between adults and nymphs. With further exploration of these data and methods in the future, even if only forewing or body and nymphal fossils of Cicadoidea are discovered, it might be possible to employ morphospace methods to individually compare these results with known fossils. Such comparisons could determine the position of known fossils on the phylogenetic tree, and clarify their relative phylogenetic relationships and evolutionary trends.

## Behaviours of communication, digging, feeding, and attendant ecological implications

In this study, we discovered tymbal structures in Cicadoidea fossils (Fig. 2q–ad). These paired, chitinous membranous exoskeletal structures generally are thought to be associated with production of vibroacoustic signals for communication by cicadas and many of their relatives (Hemiptera: Cicadomorpha). We found tymbal structures (Fig. 2q–x) in the segmented data from micro-CT scans of the male fossils of *E. emeryi* gen. et sp. nov., *C. problematica* comb. nov., *C. shcherbakovi* sp. nov., *Pr. xiai* gen. et sp. nov., and the female of *Pr. xiai* gen. et sp. nov., as well as a developed tymbal muscle structure in the male *Pr. xiai* gen. et sp. nov. (Fig. 2y–ad).

The presence of a tymbal structure in all stem groups suggests that it represents a plesiomorphic trait of Cicadoidea, and one that both males and females of Cicadoidea had in their early evolution. Compared to micro-CT data, we found that the outline of the ancient tymbal organ seems to be rounder than those of extant species. Additionally, a portion of the ancient tymbal appears crinkled (Fig. 2q, t), which may be caused by a muscle pulling on the tymbal which consists of sclerotised cuticle and resilin arranged into a ribbed structure[58]. Other features linked with sound and auditory structures, such as the tymbal cavity, tympanum, tympanal cavity, and operculum, which typically are reasonably developed in the majority of Cicadidae

species, were not detected in these fossils (Fig. 2q–ak). Therefore, these mid-Cretaceous fossils probably did not have the ability to exploit the interaction of these specialised structures to produce the high-decibel songs of the crown cicadids. The mid-Cretaceous taxa probably were similar to modern tettigarctids, which communicate through vibrational signals transmitted through the substrate[14]. However, the fossil *Pr. xiai* gen. nov. sp. nov. was found to have preserved a well-developed tymbal muscle and a cavity within its abdomen (Fig. 2aa, ab), similar to the male abdomen of extant Cicadidae instead of Tettigarctidae which has undeveloped tymbal muscles and no resonant abdominal cavity[14] (Supplementary Fig. 18). As the internal organs, malpighian tubules, and muscles of *Pr. xiai* gen. nov. sp. nov. were preserved (Supplementary Fig. 7), we conclude that the abdominal cavity was not hollow due to decomposition and loss of internal components but existed premortem. Sound production behaviour of *P. xiai* gen. nov. sp. nov. may have been comparable to that of modern Cicadidae, in that the movement of the tymbal muscles in the cavity pulled the tymbal membrane back and forth, causing the inward buckling of the ribs that produce the tymbal membrane to create sound[11–13]. The abdominal cavity of *Pr. xiai* gen. nov. sp. nov. may have had a resonant function. This explanation also seems to support the hypothesis that some taxa had started to produce sounds louder than typical tettigarctid substrate vibrations during the mid-Cretaceous. In any case, species of Cicadoidea may have been relatively silent for most of the duration of the Mesozoic compared with modern singing cicadas. It will be necessary to look for auditory organs in more fossil samples and from sources other than Kachin amber to test whether other modes of vocalisation existed during this time. The discovery of tymbals and tymbal muscles in Kachin amber cicadas suggests that the potential for preservation of other key evolutionary structures in different taxa within Kachin amber fossils.

In this study, we also report the a final instar nymph and the exuviae of cicadoid fossils from mid-Cretaceous (Fig. 1i–m). The most conspicuous characteristic of the fossil final instar nymph and exuviae is that they possess powerful and modified forelegs which are comparable to current cicadoid nymphs (Supplementary Fig. 15). This feature also is an important identifying character for other cicada nymph fossils, from the first to the final instar.The possession of distinctive forelegs in the fossil nymphs suggests life habits similar to those of modern cicada nymphs. These fossil nymphs exhibit a sickle-shaped tibia of the enlarged foreleg that fits against the expanded femur, forming a grasping configuration, indicating a strong capability for soil excavation and transportation. Current evidence shows that cicada nymphs can live underground up to 3.66 m in depth[59]. Such nymphs use their specialised forelegs to dig tunnels for food, achieve mobility in the soil, and create earthen chambers near plant roots[20,60,61]. Cicada nymphal tunnels can be recognised from meniscate sediment backfill, and from pan-shaped chamber trace fossils[20,61]. The specialised foreleg of these fossil nymphs allows a strong inference regarding their digging behaviour and subterranean lifestyle.

The lifestyle of nymphs living underground for years on end can reduce their competition for aboveground nutrients and provide refuge from harsh surface conditions. Our fossils show that cicada nymphs occupied this novel ecological niche by, or at least, during the mid-Cretaceous (Fig. 4). Given that Kachin nymphal fossils have specialised forelegs that closely resemble their counterparts in modern cicadas, we surmise that the enlarged foreleg, associated fossorial behaviour, and subterranean life habit originated substantially earlier than mid-Cretaceous. Although no evidence of cicadoid nymphs has been found in Middle Jurassic (Callovian) Daohugou beds—all Daohugou cicadoids are adults—it is likely that these taxa had similar lifecycles to modern cicadas that included nymphs with fossorial habits and an underground lifestyle. Fossils found in Kachin amber and a mass emergence of adult cicadas from the Jurassic Daohugou Lagerstatte demonstrate that they emerged from the soil and ascended

upward to an arboreal habitat[33,62], implying considerable niche differentiation and a substantial biomass transfer from belowground to above-ground habitats in the mid Mesozoic by their lifecycles.

Ecosystem changes caused by cicada lifecycle activities excellently illustrate the ecological concept of 'resource pulse'[15,21]. Different stages of the cicada life cycle, including nymphal root-feeding, emergence from soil, adult feeding on xylary sap, distinctive adult oviposition into woody tissues, and adult death can significantly affect the growth of their host plants, the number of predators, and the nutrient cycle of local food webs and ecosystems[21–23,25,63]. Based on the presence of rich records and varied adult, nymphal and exuvial cicadoids found during the mid-Mesozoic and their similarity to morphologies and behaviours with extant counterparts, it is plausible that a similar ecological pattern existed among Cicadoidea in mid-Mesozoic plant communities. As in extant cicadas, Mesozoic cicadas likely created a resource pulse to some extent, and thus may have played an important role in the nutrient recycling of mid-Mesozoic ecosystems.

The ecological shift in spatial niches of cicada nymphs versus adults is accompanied by the transition from underground root feeding to aboveground stem feeding during the process of niche shift. Since root-feeding behaviour generally occurs in an enclosed (typically subterranean) habitats, its traces are not readily preserved in the fossil record. The evidence for root-feeding is the most poorly documented herbivore functional feeding group in the fossil record[64]. Fossil evidence among ancient terrestrial arthropods mainly includes borings and fossilised faeces left behind in root fossils from the late Pennsylvanian to Late Triassic[65] (Supplementary Data 4 for a summary of the fossil records of arthropod root-feeding). Cicada nymph fossils with fossorial forelegs here provide one of the few, explicit fossil records of root-feeding behaviour. Root feeding undoubtedly provides advantages for cicada nymphs to migrate to and to inhabit for extended time underground, apparently representing a successful and highly specialised survival strategy.

Cicadas feed on xylem sap of plant vascular tissue. They use their stylets to enter surface and deeper-seated tissues and terminate at vessel to suck xylem sap[66–69]. The suctioning of xylem requires use of strong cibarial pumping muscles and overcome xylem's negative tension; the cibarial pump apparatus is reflected in the substantially enlarged clypeus with distinct transverse or chevron-shaped muscle insertion grooves[70,71]. The similarity of the cibarial apparatus structure between nymphs and adults of fossil and modern taxa suggests that the fossils had a similar ability to feed on xylem sap (Fig. 2i–p and Supplementary Figs. 12e, 13d). Xylem sap–feeding in Hemiptera is thought to have a single origin in the Clypeata[72,73], at least based on the fossil evidence[74–76]. We infer that those adaptive behaviours associated with feeding on xylem had been acquired at the origin of Cicadoidea.

Extant cicadas feed on angiosperms, with a few examples of feeding records on gymnosperms[77,78]. Early instar nymphs can feed on tiny plant roots, whether gymnosperm or angiosperm, based on field experiments[67]. Given the feeding needs of a large number of Jurassic cicadoids from Daohugou[33,61], and considering gymnosperm dominance in the Jurassic Daohugou forest[79], gymnosperms most likely provided significant quantities of food to cicadas during that time interval. We consider that there might be a broad host shift or tendency in the evolution of Cicadoidea to feed on angiosperms when this newly emerged plant group diversified during the Early Cretaceous. During the mid-Cretaceous, when the resin that originated the Kachin amber deposits was produced and accumulated, many typical angiosperms had already appeared, and woodland and forests dominated by angiosperms were emerging[80–83]. It is possible that around this time, cicadoids underwent a transition period involving a major shift of their host plants.

However, the reasons why cicadoids were on angiosperms remains to be considered. Are angiosperms providing more favourable

host conditions than gymnosperms? In other words, is it random selection of plant hosts, or is the relationship one of obligatory adaptation? It cannot be overlooked that microbial symbionts play an indispensable role in assisting these herbivorous insects in adapting to host plants and nutrient provisioning[84–86], while studying this relationship in fossil research remains challenging. Our morphological analyses indicate that Mesozoic cicadoids seemed to have occupied a special morphospace than its modern relatives (Fig. 3b–d). The development of angiosperms does not appear to have aided the Cicadoidea to generate greater morphological diversity. Compared to their Mesozoic progenitors modern Cicadoidea have a distinct widening of the head and of the postclypeus with the emergence of central sulcus (Supplementary Fig. 10 and Fig. 2a–p), which may be the direct reflection of the changes of the host plant resistance from tissues to gain efficiencies in feeding capacity. Changes in head width are accompanied by further changes in the thorax (Fig. 3b–c), reflected in the thorax notum, along with changes in the wings and their venation, which may indicate shifts in flight muscles and flight ability, as well as the change in forest spatial structure caused by the transition from a gymnosperm- to angiosperm-dominated community composition. The general broadening of the head and thorax in Cicadidae are accompanied by the tendency of abdominal widening and muscular modifications, which may be an important factor influencing the subsequent development of tymbal organs in singing cicadas, and subsequently underwent great radiation evolution in the Cenozoic period.

## Methods

### Materials

The amber specimens, including six new adults and five nymphs, described here originated from the Hukawng Valley in Tanaing Township, Myitkyina District of Kachin State, in Myanmar. The geographical coordinates of the site are ~26°15′N, 96°34′E. The age of these ambers is 98.79 ± 0.62 Ma or early Cenomanian of the earliest Late Cretaceous[87]. Specimens MGM2016–014 and MGM2016–017 are deposited in the Myanmar Gems Museum in Nay Pyi Taw, Myanmar. Specimens NIGP201895, NIGP201896, NIGP201897, NIGP201898, NIGP201900 and NIGP201901 are deposited in the Nanjing Institute of Geology and Palaeontology, Chinese Academy of Sciences (NIGPAS), in Nanjing, China. Specimens LYU–BC2001, LYU–BC2002, LYU–BC2003 and LYU–BC2004 are deposited in the Linyi University, Linyi, China. Amber specimens were collected in 2013–2016, before the Myanmar army closed the Kachin amber mines in November 2017. The fossils were acquired in full compliance with the laws of Myanmar. All authors declare that the ambers reported in this study are not involved in armed conflict and ethnic strife in Myanmar.

The specimens of extant *Tettigarcta* were collected in Tasmania and Victoria, Australia and provided by Prof. David Emery from University of Sydney. The specimens of extant *Hyalessa maculaticollis, Platypleura kaempferi* and *Cryptotympana atrata* were collected from Nanjing, Jiangsu Province, China. The descriptive terminology for the adult structures is mainly based on refs. 1,8; and for the nymphal structures, the descriptive terminology is mainly based on ref. 88 and follow the changed terminology of forewings in this study (Supplementary Note 5 and Supplementary Fig. 19) based on new anatomical observations and a reconciled terminology. The most significant alterations are the discovery of a short RA$_1$ vein of the forewing near the nodal line. The structure name, point of blade of tibia (pbt), is changed to secondary apical tooth of tibia (sapt). For other interpretations please see Supplementary Note 4.

### Optical microscopy

Photographs were taken using a Zeiss AXIO Zoom V16 stereo microscope system at the State Key Laboratory of Paleobiology and Stratigraphy, NIGPAS, and at the Institute of Geosciences, University of Bonn, Germany, and using a Keyence VHX-7000 microscope at Institute of Geology and Paleontology, Charles University, Czech Republic. Each image was digitally stacked with 10–60 individual focal planes, using the software Helicon Focus 7 for better illustration of the 3-dimensional (3D) characteristics.

### X-ray micro-computed tomography (Micro-CT)

Specimens MGM2016–017, NIGP201896, NIGP201897, LYU–BC2001, *T. crinita* (male), *T. crinita* (female), *Pl. kaempferi* (male), *Pl. kaempferi* (female) were scanned using a 3D X-ray microscope (3D-XRM) (Zeiss Xradia 520 Versa) at the State Key Laboratory of Paleobiology and Stratigraphy, NIGPAS. Images were generated using CCD-based 0.4× and 4× (only LYU–BC2001) objectives, at an X-ray voltage of 50 kV (power 4 W), with voxel sizes of 16.357, 15.519, 20.317, 4.1036, 41.149, 49.601, 35.513, 45.092 μm, respectively. Specimens, LYU–BC2002, NIGP201895, and NIGP0011 were scanned using a phoenix v|tome|x s micro-CT at the Institute of Geosciences, University of Bonn. Images of LYU–BC2002, NIGP201895, and NIGP0011 were generated by a CCD-based 40x (only LYU–BC2002) and 20× objective and at an X-ray voltage of 100 kV (120 mA), 90 kV (130 mA), 80 kV (80 mA) with the voxel size 5.1, 9.9, and 9.9 μm, respectively. Data for all specimens were collected over 360°, and the number of images ranged from 1301 to 2051. The exposure time for each projection is between 1.5 s and 5 s. Volume data processing was performed using the software Avizo 8.1 (FEI Visualization Sciences Group, Burlington, MA) and VGStudio Max (version 3.0, Volume Graphics, Heidelberg, Germany).

### Phylogeny

We established a multifaceted morphological scheme of phylogenetic data analyses that included Cicadoidea fossils and revealed their phylogenetic placements. We performed phylogenetic analyses under parsimony and Bayesian optimality criteria setting a data matrix of 81 characters and 22 taxa based on an established classification status close to the family level[1,73,89] (Fig. 3a and Supplementary Fig. 8, Supplementary Note 2 and Supplementary Data 1). The phylogenetic matrix was coded in the software Notepad++. For outgroups that constrained tree topology, we used Hylicellidae including *Cycloscytina gobiensis*, Cercopidae including *Aphrophora pectoralis, Cercopis vulnerata*, Procercopidae including *Jurocercopis, grandis, Anthoscytina elegans*, and *Paranthoscytina xiai*[90–93]. We chose the type genus of *Cycloscytina* for Hylicellidae (Hemiptera) as the root of the tree. The parsimony analysis was conducted in TNT (v.1.6)[94], using the Traditional Search option, with memory to store 30,000 trees, 10,000 replicates, with 100 trees to save per replication, and using the tree bisection reconnection algorithm. Four trees were obtained and the strict consensus tree was used equal weighting parsimony[95]. Bootstrap analyses were performed using a traditional search and 10,000 replicates, with outputs saved as absolute frequencies. Obtained trees were viewed using WinClada 1.00.08[96]. Supplementary Fig. 8 displayed the unambig changes only, slow optimisation and fast optimisation results of the strict consensus tree[97] in Winclada. The Bayesian analysis was conducted in MrBayes (v.3.2.7a)[98] using a Markov chain Monte Carlo statistical model. The analysis was run for 5,000,000 generations with a tree sampling frequency of every 100 iterations. Burn-in for tree samples was set at 25%. Convergence of independent runs was assessed using average standard deviation of split frequencies (about 0.0017), and potential scale reduction factors (about one for all parameters).

### Morphological disparity

We selected ten landmarks to quantify morphological characteristics (Fig. 3b, Supplementary Fig. 9 and Supplementary Tables 1–3) to do the the principal components analysis (PCA) of geometric morphometrics on thorax and head profiles in dorsal view. We included 55 genera, with 4 fossil genera and 51 extant genera, resulting in a dataset

of 175 specimens for analysis (Source Data). For the analysis of non-metric multidimensional scaling (NMDS) of forewing, we employed 32 characteristics of forewings of 90 species from 69 genera of cicadoids (Fig. 3c; Supplementary Note 3 and Supplementary Data 2). For the PCA of forewing profile, we selected nine landmarks to quantified the forewing profile. We included 66 genera, with 30 fossil genera and 36 extant genera, resulting in a dataset of 238 specimens for analysis (Supplementary Figs. 10a, 11, Supplementary Tables 4–6, and Source Data). We selected nine characters of the nymphal forelegs of seven species including five fossils and two extant species represented by extant Tettigarctidae and Cicadidae for the principal co-ordinates analysis (PCoA) (Fig. 3d and Supplementary Tables 7, 8). Landmarks were collected by tpsDig232. The PCA of the landmark data from the dorsal profile was standardised in Past 3.15 by Procrustes analysis and then conducted in OriginPro 2021. NMDS analysis of the characteristics of forewings was conducted in Past 3.15 selected Raup-Crick similarity index. PCA of the landmarks of the forewing profile, and the PCoA of the characters of nymphal forelegs were conducted in Past 3.15[99]. The morphometric measurement data of cicadoid adults was collected by Image J[100] (Supplementary Fig. 10b–f), mainly based on previous studies and other internet resources (please see in Source Data). Genera in all datasets involving adult specimens include specimens from both extant Tettigarctidae and the five subfamilies of Cicadidae.

## Figures preparation and artistic reconstruction
Figures were prepared in Adobe Photoshop CS6 and CorelDraw 2023. Line drawings were prepared in Procreate. The artistic reconstruction of Fig. 4 was prepared in Pixologic ZBrush, Maya and Adobe Photoshop CC.

## Reporting summary
Further information on research design is available in the Nature Portfolio Reporting Summary linked to this article.

## Data availability
The micro-CT data generated in this study are available in the figshare database under accession code: https://figshare.com/s/149753caed044ed898c5 (https://doi.org/10.6084/m9.figshare.24715743). Please see ref. 101 Specimens MGM2016–014 and MGM2016–017 are deposited at Myanmar Gems Museum in Nay Pyi Taw, Myanmar. Specimens NIGP201895, NIGP201896, NIGP201897, NIGP201898, NIGP201900 and NIGP201901 are deposited at Nanjing Institute of Geology and Palaeontology, Chinese Academy of Sciences (NIGPAS), in Nanjing, China. Specimens LYU–BC2001, LYU–BC2002, LYU–BC2003 and LYU–BC2004 are deposited at Linyi University, Linyi, China. They are available on request. Access to the data used in this study, which includes previously published data or data available through online repositories, and the data generated in this study, is provided in the Supplementary Information, Supplementary Data, and Source Data files. Source data are provided with this paper.

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

## Acknowledgements

We thank Dinghua Yang for the creation of the artistic reconstructions; Suping Wu for help with the micro-computed tomography scan; David C. Marshall, Torsten Wappler, Changtai Niu for their constructive comments and discussions; Zeyuan Sun, Xianye Zhao, Yilun Yu, Lucia Šmídová, Yanzeh Fu, Naihua Xue and Hantao Hui for some helpful discussions of methods, instrument and specimens; Boris Chauviré for providing micro-CT data of nymphal fossil from Pliocene opal. This research was supported by the National Natural Science Foundation of China (Nos. 42125201, 42172033, 42288201 to B.W., H.Z.); Deutsche Forschungsgemeinschaft (DFG) (No. 53 to J.R., B.M.) and as a contribution of the DFG research unit FOR2685 'The Limits of the Fossil Record'; Natural Scientific Foundation of Shandong Province (ZR2020YQ27 to J.C.); Cooperation project of Charles University (H.J.); State Key Laboratory of Palaeobiology and Stratigraphy (No. 223114 to H.J.); CAS-DAAD Fellowship Programme (H.J.); and the President's International Fellowship Initiative (PIFI) supported by Chinese Academy of Sciences (J.S.). This work contributes to the Deep-time Digital Earth (DDE) Big Science Programme.

## Author contributions

B.W. H.Z. and H.J. designed the project; H.J. collected data and conducted the analysis with help from J.S., C.C.L., M.S.M., B.W., J.C., A.D.M., B.M., C.W., H.Z., J.R. Z.D. T.T.N.; B.M. conducted the micro-CT scanning of a subset of the specimens; H.J. segmented the microtomographic data with help from B.M.; H.J. wrote the manuscript and made the figures with help from C.C.L., J.S., J.C., M.S.M., B.W., A.D.M., H.Z., J.R. All authors commented on the manuscript.

## Competing interests

The authors declare no competing interests.
