## [Peer Review File · Nature Communications]

Mesozoic evolution of cicadas and their origins of vocalization and root feedingReviewers' Comments:

Reviewer #1:

Remarks to the Author:

The manuscript presents the first report of the presence of tymbal and tymbal muscles in a cicadoid fossil, and the presence of fossorial forelegs in a final-instar nymph fossil, allowing to suggest communication, digging and feeding behaviours in stem cicadoids. Still, the authors performed a phylogenetic analysis based on morphology to test the position of new stem genera and species and the relationship between them and crown cicadoids selected species; they used characters from geometric morphometrics analysis and Non-metric Multidimensional Scaling (NMDS) to compare the morphology of head and notum, forelegs and wings venation between stem and crown cicadoids, and finally they performed micro-computed tomographic scans to show a Cicadoidea morphological evolution.

The introduction is adequate with a good bibliographic review including recent papers. The methods adopted are suitable for the objectives of the paper, as mentioned before. The supplementary material is rich in information including figures and tables. My only suggestion for this section is to include the files of the character matrix and phylogenetic analysis.

The results and discussion sections have sentences wrongly allocated. I missed a better explanation of some figures in the results section, and I think the discussion section has phrases that should be in the results section.

Below are some suggestions and comments to improve the manuscript:

68. I suggest changing "vocal expression" to "sounds expression" or "sounds communications" because the term "vocal" refers to having a voice.

96. "Currently, all Mesozoic fossils are classified as Tettigarctidae, based principally on forewing venation", but in Moulds (2018) table 1 (continued) page 446, *Burmamacada protera* Portinar & Kritsky, 2011 (Cicadidae) it is reported to Late Cretaceous.

105. I suggest you report in this paragraph the analysis you performed (phylogenetic, geometric morphometrics, NMDS) and for what purpose.

122, 217. If you are establishing a new genus and this one as a type genus, you have to determine the family-group level which this new genus belongs to and is the type, e.g. family, subfamily or tribe. For example, *Cicadetta* is the type genus of Cicadettinae (subfamily); *Fidicina* is the type genus of Fidicinini (tribe). Stem cicadoids is not a taxonomic name and therefore should not be considered to establish a type genus. If this determination is not possible because it is a fossil, then you should only establish as a new genus of Cicadoidea and not as a type genus. Please, see the article 64 of International Code on Zoological Nomenclature.

114. I know the manuscript can be limited by length to be accepted in this journal, but my suggestion is, if possible, to include the full description of new genus and species in the main text, and not in supplementary material. The taxonomy is an important part of this study, and it is not practical for a reader see the descriptions truncated.

133. I didn't find the diagnosis of *Eunotalia* gen. nov. and *E. emeryi* sp. nov. in the main text or supplementary notes.

150. I suggest adding a section name "included species" for each genus and list there the species included.

213. *Vetuprosbole parallelica* isn't in italic.

256. ..." nymphal fossils with adult fossils in a future study, we provisionally categorise them". Change the point for a comma.

265. I suggest repeating the figures related to the nymph species in this sentence.

276. I think there is a reference for this sentence missing.

312. Many parts in the discussion seem to be results. I suggest separating some sentences, e.g. "The results based on 81 characters (Fig. 3; Supplementary Fig. 13) demonstrate..." and include it in a section for Morphological analysis of adult fossils. Or change the name of the section to Results and Discussion.

313. I think it is better to specify if the 10 specimens are adults.

322. I suggest referring to the 81 characters with the Supplementary Note 4 together with the figures.

334. "previous taxonomic treatments" is relative to stem or crown cicadoids?

337. Fig. 3b – Change the B to lower case.

391. Is this statistical analysis some part of the supplementary material? If so, it should be referenced here.

400. "The distal aspect of the mesonotum in crown cicadids has a cruciform elevation, whereas the rest of cicadoid fossils have scutellum" I don't understand this sentence since cruciform elevation and scutellum are synonyms (Moulds 2005, fig. 3, page 378).

454. I didn't understand the reference to Supplementary Figure 19 here.

583. It is usual to mention the full name of the taxon at the first mention in the text.

617. It would be interesting to mention the software where the characters were coded.

623. The root should be stated.

Phylogeny - I missed a better explanation of the results of the phylogenetic analysis and the characters in the final tree, e.g. which of the characters were recovered as synapomorphies and homoplastic? I didn't find the results relating to the number of recovered trees and an explanation of how the implied weight analysis was performed as well as the calculation of k-values (e.g. SPR similarity) and which k-values were recovered.

Supplementary Figure 19 – the meaning of the abbreviation "tcav" is missing in the caption.

Figure 2. I suggest putting on the side of each species' name the correspondent letter, furthermore the species referent to the letter h is missing in the caption.

Figure 3a. When performing a phylogenetic analysis, it is not usual to use terminals with different ranks. I suggest changing within this figure the terminals represented by genera and subfamilies to species. Moreover, I didn't find the information of which species were used representing the genera and subfamilies in the analysis, only some of them in the Supplementary Table 2, and in the lines 583 and 600. Besides, the meaning of the numbers in the internal nodes is missing to be informed in the caption. These numbers are not mentioned in the main text (results or discussion).

Fig. 3b, c, d. I missed a better explanation of what was found as results and what these figures represent.

Tatiana P. Ruschel

Reviewer #2:

Remarks to the Author:

The current contribution describes new adult and nymphal cicadoid insects from mid-Cretaceous

Kachin amber in Myanmar, achieving a great detail of morphological understanding, presents a phylogeny of the group using both fossil and extant taxa, and provides a series of morphospace analyses using abundant data. The results significantly enrich the knowledge on the evolution of the group, identify tymbal structures and other associated structures such as muscles and a probably resonating cavity for the first time in the fossil record of these insects, allowing to make inferences about their possible sound production ability and communication. The nymphal cicadeoid fossils described —previously only known by an early-instar nymph also from Kachin amber and from a final-instar nymph drawn (yet not described) from slightly younger New Jersey amber (these works cited by the authors in Results), both with fossorial-type forelegs— have allowed the authors to infer and discuss the likely digging and underground habits of these immature insects by comparing with their modern counterparts.

Although the text is generally well-written and makes a compelling case, it contains several language elements throughout that required improvement. I have tried to mend as many of these as I could, but please make an additional effort to polish this aspect.

Taxa description and analyses are thorough and detailed. Figures are of excellent quality.

Please note that than I'm not an hemipterologist nor a native English speaker.

I would like to highlight a few matters that I think are necessary to tackle before the present manuscript can be considered for publication. These, plus the language issues noted above, justify my decision of MINOR-MODERATE reviews for the present work.

1) Need to tone down some claims by adding the proper degree of uncertainty.

First, I would argue that the finding of nymphs with fossorial legs does not represent direct evidence of their habit and behaviour, since these are inferred through morphology and actualism, which my colleagues would agree that they are not devoid of problems. Therefore, I think adding a certain degree of caution and uncertainty is in order, both in abstract (introduction of "had most likely" in line 55) and discussion ("The specialized foreleg of in these fossil nymphs allows to infer their digging behaviour and subterranean lifestyle" rather than "The specialized foreleg of in these fossil nymphs demonstrates their digging behaviour and subterranean lifestyle").

Also, line 525 reads "Cicada nymph fossils with fossorial forelegs here provide one of the few direct fossil records of root-feeding behaviour". Similarly as the case above, I would argue that, since the authors are inferring root-feeding behaviour through the same structures that modern relatives having these fossorial habits possess, the present record does not represent a direct evidence of root-feeding behaviour but an indirect one.

Moreover, the fact that all the material comes from a single locality/region from the mid-Cretaceous limits how general and global the conclusions can be, and so this aspect must be treated with caution. The most clear case of this is shown in the sentence around line 470 "species of Cicadoidea were comparatively silent for of* the Mesozoic compared with modern singing cicadas" (see context), which in my opinion has to be changed to "species of Cicadoidea may have been relatively silent during the Mesozoic compared with modern singing cicadas".#

Other instances in which the need to increase the degree of caution/uncertainty in the author's statements is necessary from my point of view are provided in the annotated manuscript.

2) Need to revise the format of the systematic section.

I understand that a effort of condensing text and information has been made in order to save space in the 'Systematic palaeontology' section of Results, but in the process, there have been several

alterations that affect the understandability of the section and its formal structure in my opinion. "Material" and "Description" subsections should always be directly under a properly introduced species name in which specimens are then listed, not under a genus heading without proper species introduction. The use of "Type genus" and "Type species" is problematic throughout the section. A species should be introduced after the "Type species" has been mentioned under a given genus, so the "Type species" line should not replace the formal introduction of a species (these are not redundant). I would also advise against merging etymology sections of new genera and their only species within (as with *Eunotalia* and *Pranwanna*). Lastly, a diagnosis is missing for the genus *Eunotalia* or its only species.

3. One last matter that I would comment here relates to taphonomy and might impact, to a certain extent, the detailed reconstruction.

Regarding the sentence "Fossils found in Kachin amber and a mass emergence of adult cicadas from the Jurassic Daohugou Lagerstätte demonstrate that they emerged from the soil and ascended to the tree canopy^{33,50}" in line 503, the fossils in Kachin amber could have also been trapped relatively close to the ground level or at the trunk level. The insects would not have necessarily had to reach the tree canopy to get trapped in resin, based on their preservation in amber. Thus, "towards the tree canopy" seems a more suitable expression.

Moreover, roots are known to produce resin in abundance. Are there any taphonomic characteristics of the amber pieces where the cicadoids are preserved that allow to ascertain the aerial origin of the amber pieces and so that the nymphs were actually ascending to or towards the tree canopy? At least some of the amber pieces actually look cloudy due to abundant debris, which is not conclusive of "litter amber" but might be suggestive of it together with other signs (syninclusions, etc), which might be more indicative of resin entrapment close to the ground. Please consider this option.

Similarly, for the reconstruction it would be highly advisable to consider the option that fossils could have been trapped by resin fallen to the ground and capturing fauna therein ("litter amber"). Also, it is known that roots can produce resin in abundant quantities (for instance, see Alvarez-Parra et al, 2021, Dinosaur bonebed amber from an original swamp forest soil). Although the taphonomy of root resin production is poorly understood, and there is no evidence --to my knowledge-- yet of insect entrapment by subaerial resin (that produced by the roots), provided that the current reconstruction showcases the underground layer, not showing this aspect --yet tentatively-- would appear to be a missed chance due to the overall detail of the illustration.

Reviewer #3:

Remarks to the Author:

Nature communications

General comments- The authors performed micro-computed tomographic scans on 10 specimens from mid-Cretaceous Kachin amber and extant cicadoids. By combining fossil and extant taxa, and using more detailed anatomical features revealed by micro CT results and morphologically based phylogenetic analysis; forewing and nymphal foreleg, revealed morphological evolutionary tendencies in Cicadoidea. This is a very interesting, original and important approach, however due to the nature of the studied material (fossil), important issues related to the M & M must be clarified. Please check some specific points below.

Geometric morphometrics is a very sensitive technique. Unfortunately, I do not feel confident to evaluate whether the images obtained from specimens in the amber are precise enough for a geometric morphometric analysis. Any possible discrepancies in the position or in the condition of preservation of the analyzed insects could lead to misinterpretations and questionable information as basis for the new genera and species description.

The finding of the first evidence for vocalization associated tymbal structures in fossils, suggesting mid

Cretaceous and some other interesting ecological aspects based on the analysis nymphs seems very robust and inedited. However, the scientific basses for the descriptions of new species and genera could be questionable.

I presented below some brief suggestions for improving the manuscript but, unfortunately, I don't feel comfortable to either recommend or reject this manuscript for publication.

Specific comments

Abstract- Brief comments on the geographic area where the specimens where found.

Main

Lines 76-77 In the sentence "live underground for as many years and tunnel through substrate using fossorial behaviors while feeding on xylem sap of roots from the first to fifth instars" and in the Lines 79-80 " several weeks ". Terms like "many Years and several weeks" should be avoided. The following sentences in the manuscript could be synthesized including the precise information missing in the mentioned lines.

Material and Methods

If possible, please provide the geographic coordinates from the sites where the analyzed samples were found.

Please comment whether the references cited for morphological nomenclatural were the same ones used for identification and comparisons for the new samples' descriptions.

I would suggest including the in text, the ten quantifiable morphological characteristics from the body dorsal profile of cicadoid species.

It is not clear whether the wing venation of the amber images could be precisely used for a geometric morphometrics analysis. The samples images in the amber are not defined by precision for analysis.

Please explore this point for a better description of the methods

Legends- All legends must be self-explanatory. Please provide additional relevant information in the legends for all figs. For instance: geographic location, technique used, In the case of fig 1 geographic location, fig 2 the technique used for the images obtained; fig 3 number of species and specimens illustrated, please name the method used for the phylogenetic analysis etc; Fig 4 please provide basic information on the method used for the image reconstruction.

REVIEWER COMMENTS

Reviewer #1 (Remarks to the Author):

The manuscript presents the first report of the presence of tymbal and tymbal muscles in a cicadoid fossil, and the presence of fossorial forelegs in a final-instar nymph fossil, allowing to suggest communication, digging and feeding behaviours in stem cicadoids. Still, the authors performed a phylogenetic analysis based on morphology to test the position of new stem genera and species and the relationship between them and crown cicadoids selected species; they used characters from geometric morphometrics analysis and Non-metric Multidimensional Scaling (NMDS) to compare the morphology of head and notum, forelegs and wings venation between stem and crown cicadoids, and finally they performed micro-computed tomographic scans to show a Cicadoidea morphological evolution.

The introduction is adequate with a good bibliographic review including recent papers. The methods adopted are suitable for the objectives of the paper, as mentioned before. The supplementary material is rich in information including figures and tables. My only suggestion for this section is to include the files of the character matrix and phylogenetic analysis.

The results and discussion sections have sentences wrongly allocated. I missed a better explanation of some figures in the results section, and I think the discussion section has phrases that should be in the results section.

Below are some suggestions and comments to improve the manuscript:

68. I suggest changing “vocal expression” to “sounds expression” or “sounds communications” because the term “vocal” refers to having a voice.

96. “Currently, all Mesozoic fossils are classified as Tettigarctidae, based principally on forewing venation”, but in Moulds (2018) table 1 (continued) page 446, *Burmacicada protera* Portinar & Kritsky, 2011 (Cicadidae) it is reported to Late Cretaceous.

105. I suggest you report in this paragraph the analysis you performed (phylogenetic, geometric morphometrics, NMDS) and for what purpose.

122, 217. If you are establishing a new genus and this one as a type genus, you have to determine the family-group level which this new genus belongs to and is the type, e.g. family, subfamily or tribe. For example, *Cicadetta* is the type genus of Cicadettinae (subfamily); *Fidicina* is the type genus of Fidicinini (tribe). Stem cicadoids is not a taxonomic name and therefore should not be considered to establish a type genus. If this determination is not possible because it is a fossil, then you should only establish as a new genus of Cicadoidea and not as a type genus. Please, see the article 64 of International Code on Zoological Nomenclature.

114. I know the manuscript can be limited by length to be accepted in this journal, but my suggestion is, if possible, to include the full description of new genus and species in the main text, and not in supplementary material. The taxonomy is an important part of this study, and it is not practical for a reader see the descriptions truncated.

133. I didn't find the diagnosis of *Eunotalia* gen. nov. and *E. emeryi* sp. nov. in the main text or supplementary notes.

150. I suggest adding a section name “included species” for each genus and list there the species included.

213. *Vetuprosbole parallelica* isn't in italic.

256. ...” nymphal fossils with adult fossils in a future study, we provisionally categorise them”. Change the point for a comma.

265. I suggest repeating the figures related to the nymph species in this sentence.

276. I think there is a reference for this sentence missing.

312. Many parts in the discussion seem to be results. I suggest separating some sentences, e.g. “The results based on 81 characters (Fig. 3; Supplementary Fig. 13) demonstrate...” and include it in a section for Morphological analysis of adult fossils. Or change the name of the section to Results and

Discussion.

313. I think it is better to specify if the 10 specimens are adults.

322. I suggest referring to the 81 characters with the Supplementary Note 4 together with the figures.

334. "previous taxonomic treatments" is relative to stem or crown cicadoids?

337. Fig. 3b – Change the B to lower case.

391. Is this statistical analysis some part of the supplementary material? If so, it should be referenced here.

400. "The distal aspect of the mesonotum in crown cicadids has a cruciform elevation, whereas the rest of cicadoid fossils have scutellum" I don't understand this sentence since cruciform elevation and scutellum are synonyms (Moulds 2005, fig. 3, page 378).

454. I didn't understand the reference to Supplementary Figure 19 here.

583. It is usual to mention the full name of the taxon at the first mention in the text.

617. It would be interesting to mention the software where the characters were coded.

623. The root should be stated.

Phylogeny - I missed a better explanation of the results of the phylogenetic analysis and the characters in the final tree, e.g. which of the characters were recovered as synapomorphies and homoplastic? I didn't find the results relating to the number of recovered trees and an explanation of how the implied weight analysis was performed as well as the calculation of k-values (e.g. SPR similarity) and which k-values were recovered.

Supplementary Figure 19 – the meaning of the abbreviation "tcav" is missing in the caption.

Figure 2. I suggest putting on the side of each species' name the correspondent letter, furthermore the species referent to the letter h is missing in the caption.

Figure 3a. When performing a phylogenetic analysis, it is not usual to use terminals with different ranks. I suggest changing within this figure the terminals represented by genera and subfamilies to species. Moreover, I didn't find the information of which species were used representing the genera and subfamilies in the analysis, only some of them in the Supplementary Table 2, and in the lines 583 and 600. Besides, the meaning of the numbers in the internal nodes is missing to be informed in the caption. These numbers are not mentioned in the main text (results or discussion).

Fig. 3b, c, d. I missed a better explanation of what was found as results and what these figures represent.

Tatiana P. Ruschel

Reviewer #2 (Remarks to the Author):

The current contribution describes new adult and nymphal cicadoid insects from mid-Cretaceous Kachin amber in Myanmar, achieving a great detail of morphological understanding, presents a phylogeny of the group using both fossil and extant taxa, and provides a series of morphospace analyses using abundant data. The results significantly enrich the knowledge on the evolution of the group, identify tymbal structures and other associated structures such as muscles and a probably resonating cavity for the first time in the fossil record of these insects, allowing to make inferences about their possible sound production ability and communication. The nymphal cicadeoid fossils described —previously only known by an early-instar nymph also from Kachin amber and from a final-instar nymph drawn (yet not described) from slightly younger New Jersey amber (these works cited by the authors in Results), both with fossorial-type forelegs— have allowed the authors to infer and discuss the likely digging and underground habits of these immature insects by comparing with their modern counterparts.

Although the text is generally well-written and makes a compelling case, it contains several language elements throughout that required improvement. I have tried to mend as many of these as I could, but please make an additional effort to polish this aspect.

Taxa description and analyses are thorough and detailed. Figures are of excellent quality.

Please note that than I'm not an hemipterologist nor a native English speaker.

I would like to highlight a few matters that I think are necessary to tackle before the present manuscript can be considered for publication. These, plus the language issues noted above, justify my decision of MINOR-MODERATE reviews for the present work.

1) Need to tone down some claims by adding the proper degree of uncertainty.

First, I would argue that the finding of nymphs with fossorial legs does not represent direct evidence of their habit and behaviour, since these are inferred through morphology and actualism, which my colleagues would agree that they are not devoid of problems. Therefore, I think adding a certain degree of caution and uncertainty is in order, both in abstract (introduction of "had most likely" in line 55) and discussion ("The specialized foreleg of in these fossil nymphs allows to infer their digging behaviour and subterranean lifestyle" rather than "The specialized foreleg of in these fossil nymphs demonstrates their digging behaviour and subterranean lifestyle").

Also, line 525 reads "Cicada nymph fossils with fossorial forelegs here provide one of the few direct fossil records of root-feeding behaviour". Similarly as the case above, I would argue that, since the authors are inferring root-feeding behaviour through the same structures that modern relatives having these fossorial habits possess, the present record does not represent a direct evidence of root-feeding behaviour but an indirect one.

Moreover, the fact that all the material comes from a single locality/region from the mid-Cretaceous limits how general and global the conclusions can be, and so this aspect must be treated with caution. The most clear case of this is shown in the sentence around line 470 "species of Cicadoidea were comparatively silent for of* the Mesozoic compared with modern singing cicadas" (see context), which in my opinion has to be changed to "species of Cicadoidea may have been relatively silent during the Mesozoic compared with modern singing cicadas".#

Other instances in which the need to increase the degree of caution/uncertainty in the author's statements is necessary from my point of view are provided in the annotated manuscript.

2) Need to revise the format of the systematic section.

I understand that a effort of condensing text and information has been made in order to save space in the 'Systematic palaeontology' section of Results, but in the process, there have been several alterations that affect the understandability of the section and its formal structure in my opinion. "Material" and "Description" subsections should always be directly under a properly introduced species name in which specimens are then listed, not under a genus heading without proper species introduction. The use of "Type genus" and "Type species" is problematic throughout the section. A species should be introduced after the "Type species" has been mentioned under a given genus, so the "Type species" line should not replace the formal introduction of a species (these are not redundant). I would also advise against merging etymology sections of new genera and their only species within (as with Eunotalia and Pranwanna). Lastly, a diagnosis is missing for the genus Eunotalia or its only species.

3. One last matter that I would comment here relates to taphonomy and might impact, to a certain extent, the detailed reconstruction.

Regarding the sentence "Fossils found in Kachin amber and a mass emergence of adult cicadas from the Jurassic Daohugou Lagerstätte demonstrate that they emerged from the soil and ascended to the tree canopy^{33,50}" in line 503, the fossils in Kachin amber could have also been trapped relatively close to the ground level or at the trunk level. The insects would not have necessarily had to reach the tree canopy to get trapped in resin, based on their preservation in amber. Thus, "towards the tree canopy" seems a more suitable expression.

Moreover, roots are known to produce resin in abundance. Are there any taphonomic characteristics of the amber pieces where the cicadoids are preserved that allow to ascertain the aerial origin of the amber pieces and so that the nymphs were actually ascending to or towards the tree canopy? At least some of the amber pieces actually look cloudy due to abundant debris, which is not conclusive of

“litter amber” but might be suggestive of it together with other signs (syninclusions, etc), which might be more indicative of resin entrapment close to the ground. Please consider this option.

Similarly, for the reconstruction it would be highly advisable to consider the option that fossils could have been trapped by resin fallen to the ground and capturing fauna therein (“litter amber”). Also, it is known that roots can produce resin in abundant quantities (for instance, see Alvarez-Parra et al, 2021, Dinosaur bonebed amber from an original swamp forest soil). Although the taphonomy of root resin production is poorly understood, and there is no evidence --to my knowledge-- yet of insect entrapment by subaerial resin (that produced by the roots), provided that the current reconstruction showcases the underground layer, not showing this aspect --yet tentatively-- would appear to be a missed chance due to the overall detail of the illustration.

Reviewer #3 (Remarks to the Author):

Nature communications

General comments- The authors performed micro-computed tomographic scans on 10 specimens from mid-Cretaceous Kachin amber and extant cicadoidea. By combining fossil and extant taxa, and using more detailed anatomical features revealed by micro CT results and morphologically based phylogenetic analysis; forewing and nymphal foreleg, revealed morphological evolutionary tendencies in Cicadoidea. This is a very interesting, original and important approach, however due to the nature of the studied material (fossil), important issues related to the M & M must be clarified. Please check some specific points below.

Geometric morphometrics is a very sensitive technique. Unfortunately, I do not feel confident to evaluate whether the images obtained from specimens in the amber are precise enough for a geometric morphometric analysis. Any possible discrepancies in the position or in the condition of preservation of the analyzed insects could lead to misinterpretations and questionable information as basis for the new genera and species description.

The finding of the first evidence for vocalization associated tymbal structures in fossils, suggesting mid Cretaceous and some other interesting ecological aspects based on the analysis nymphs seems very robust and interesting. However, the scientific bases for the descriptions of new species and genera could be questionable.

I presented below some brief suggestions for improving the manuscript but, unfortunately, I don't feel comfortable to either recommend or reject this manuscript for publication.

Specific comments

Abstract- Brief comments on the geographic area where the specimens were found.

Main

Lines 76-77 In the sentence “live underground for as many years and tunnel through substrate using fossorial behaviors while feeding on xylem sap of roots from the first to fifth instars” and in the Lines 79-80 “several weeks “. Terms like “many Years and several weeks” should be avoided. The following sentences in the manuscript could be synthesized including the precise information missing in the mentioned lines.

Material and Methods

If possible, please provide the geographic coordinates from the sites where the analyzed samples were found.

Please comment whether the references cited for morphological nomenclature were the same ones used for identification and comparisons for the new samples' descriptions.

I would suggest including in the text, the ten quantifiable morphological characteristics from the body dorsal profile of cicadoidea species.

It is not clear whether the wing venation of the amber images could be precisely used for a geometric morphometrics analysis. The samples images in the amber are not defined by precision for analysis. Please explore this point for a better description of the methods

Legends- All legends must be self-explanatory. Please provide additional relevant information in the legends for all figs. For instance: geographic location, technique used, In the case of fig 1 geographic location, fig 2 the technique used for the images obtained; fig 3 number of species and specimens illustrated, please name the method used for the phylogenetic analysis etc; Fig 4 please provide basic information on the method used for the image reconstruction.

REVIEWER COMMENTS

REVIEWER #1

The manuscript presents the first report of the presence of tymbal and tymbal muscles in a cicadoid fossil, and the presence of fossorial forelegs in a final-instar nymph fossil, allowing to suggest communication, digging and feeding behaviours in stem cicadoids. Still, the authors performed a phylogenetic analysis based on morphology to test the position of new stem genera and species and the relationship between them and crown cicadoids selected species; they used characters from geometric morphometrics analysis and Non-metric Multidimensional Scaling (NMDS) to compare the morphology of head and notum, forelegs and wings venation between stem and crown cicadoids, and finally they performed micro-computed tomographic scans to show a Cicadoidea morphological evolution.

The introduction is adequate with a good bibliographic review including recent papers. The methods adopted are suitable for the objectives of the paper, as mentioned before. The supplementary material is rich in information including figures and tables. My only suggestion for this section is to include the files of the character matrix and phylogenetic analysis.

The results and discussion sections have sentences wrongly allocated. I missed a better explanation of some figures in the results section, and I think the discussion section has phrases that should be in the results section.

Response: Thank you very much for your valuable feedback. We have carefully considered your comments and have reorganized some of the results and discussion. Please see the detailed answer below.

Below are some suggestions and comments to improve the manuscript:

Line 68. I suggest changing “vocal expression” to “sounds expression” or “sounds communications” because the term “vocal” refers to having a voice.

Response: Agreed. Text modified the “vocal expressions” to “sounds expressions”.

Line 96. Currently, all Mesozoic fossils are classified as Tettigarctidae, based principally on forewing venation, but in Moulds (2018) table 1 (continued) page 446, *Burmacicada protera* Portinar & Kritsky, 2011 (Cicadidae) it is reported to Late Cretaceous.

Response: Thank you very much for pointing out this issue. Here we add “adult” between all and Mesozoic fossils, and modified this sentence and changed to “Currently, all adult Mesozoic cicadoid fossils are...”. *Burmacicada protera* Portinar & Kritsky, 2011 is thought to be a first-instar nymphal fossil and classified into Cicadidae. Please allow us to emphasize one additional point here. We think *Burmacicada protera* Portinar & Kritsky, 2011 as the nymphal fossil was classified into Cicadidae may be problematic. We compared the morphological structures of all nymphal fossil records with those of typical extant nymphs from the living species of Cicadidae and Tettigarctidae, and obtained the following results and conclusions “ Because both living Cicadidae and Tettigarctidae possess the accessory femur tooth (Supplementary Fig. 10q–t), its not a reliable characteristic to classify fossils into these two families. As a result, using this trait for classification makes it unreliable to place the earlier instar nymphs found in Kachin amber³¹, late Eocene Baltic amber²⁹, and mid-Miocene Dominican amber³¹ within the crown cicadid clade.”. Please see line 426-431.

Line 105. I suggest you report in this paragraph the analysis you performed (phylogenetic, geometric morphometrics, NMDS) and for what purpose.

Response: Thank you very much for your suggestion. We add the contents “We conducted a study combining morphological characters of fossils and extant taxa of Cicadoidea to perform phylogenetic analysis and morphological disparity analysis for the first time. Our study revealed clearer phylogenetic relationships between fossil and extant groups of cicadoids, and shed light on the macroevolutionary of adaptive transformations in different body structures.” in this paragraph.

Lines 122, 217. If you are establishing a new genus and this one as a type genus, you have to determine the family-group level which this new genus belongs to and is the type, e.g. family, subfamily or tribe. For example, Cicadetta is the type genus of Cicadettinae (subfamily); Fidicina is the type genus of Fidicinini (tribe). Stem cicadoids is not a taxonomic name and therefore should not be considered to establish a type genus. If this determination is not possible because it is a fossil, then you should only establish as a new genus of Cicadoidea and not as a type genus. Please, see the article 64 of International Code on Zoological Nomenclature.

Response: Thank you very much for pointing this out. We modified the establishment of the type

genera to only establish the new genera in this study. We made modifications to lines 131 and 232, revising them to *Eunotalia* gen. nov. and *Pranwanna* gen. nov.

Line 114. I know the manuscript can be limited by length to be accepted in this journal, but my suggestion is, if possible, to include the full description of new genus and species in the main text, and not in supplementary material. The taxonomy is an important part of this study, and it is not practical for a reader see the descriptions truncated.

Response: Thank you very much for your suggestion. We agree that the comprehensive description of the new genus and species is very important. However, compared to show complete description of the new genus and species based on the limited space, we would like to more emphasize the results obtained through the taxonomic work to reveal the phylogenetic relationships with extant taxa, key morphological macroevolution, and ecological behavioral analysis of fossils in this paper. All biological features that received relatively higher attention are mentioned in the main text and presented in the figures. Considering the different emphasis in research presentation, we have chosen to retain the complete description in the supplementary information for this study. We recognize the indispensable role of taxonomic work and specimen descriptions as the foundation for discussing morphological evolution, particularly in palaeontological research. Therefore, in our forthcoming study, we will further emphasize and showcase the taxonomic contents.

Line 133. I didn't find the diagnosis of *Eunotalia* gen. nov. and *E. emeryi* sp. nov. in the main text or supplementary notes.

Response: Thank you very much for pointing this out. We modified and added the contents to the section of diagnosis of *Eunotalia* gen. nov. and *E. emeryi* sp. nov. Initially, considering the word count and the fact that the specimen represents a single genus and species, we kept the key diagnostic features and comparisons in the "Remarks" section.

Line 150 I suggest adding a section name "included species" for each genus and list there the species included.

Response: Thank you very much for your suggestion. We add included species of *Cretotettigarcta*. Please see line 179-181 "The remaining fossil genera reported in this article consist of only one species each, which we have chosen not to add for now."

Line 213. *Vetuprosbole parallelica* isn't in italic.

Response: Text modified. Thanks.

Line 256 ..."nymphal fossils with adult fossils in a future study, we provisionally categorise them". Change the point for a comma.

Response: Text modified. Thanks.

Line 265. I suggest repeating the figures related to the nymph species in this sentence.

Response: Text modified. Thanks.

Line 276. I think there is a reference for this sentence missing.

Response: Thank you for pointing this out. Text modified to "the final-instar nymph of extant Cicadidae display three to nine teeth of the femoral comb⁴⁹ (Supplementary Table 1)". We have added an article and supplementary information, which presents more of the comparative results we have summarized.

49. Hou, Z., Li, Q., & Wei, C. Morphology and identification of the final instar nymphs of three cicadas (Hemiptera, Cicadidae) in Guanzhong Plain, China based on comparative morphometrics. *ZooKeys* 425, 33–50 (2014).

Line 312. Many parts in the discussion seem to be results. I suggest separating some sentences, e.g. 'The results based on 81 characters (Fig. 3; Supplementary Fig. 13) demonstrate...' and include it in a section for Morphological analysis of adult fossils. Or change the name of the section to Results and Discussion.

Response: Thank you very much for your suggestions. We agree with your suggestion to place this part of the content in both the results and discussion sections, respectively. Please see "Result" section 1.2 Phylogeny and morphological disparity and "Discussion" section 1. 1. Phylogenetic relationships, estimation of divergence time and morphological macroevolution.

Line 313. I think it is better to specify if the 10 specimens are adults.

Response: Text modified to "...and supplemented this with micro-CT scan anatomical data from six

newly reported adult amber fossils and four extant specimens included in this study.". Please lines 271-274. Thanks.

Line 322. I suggest referring to the 81 characters with the Supplementary Note 4 together with the figures.

Response: Text modified to "This examination resulted in a total of 81 morphological data features that were used to reconstruct the phylogenetic tree (Supplementary Note 4, Table 2 and Fig. 13)". Please lines 274-276. Thanks.

Line 334. previous taxonomic treatments' is relative to stem or crown cicadoids?

Response "previous taxonomic treatments" refers to that all adult Mesozoic fossils are classified as Tettigarctidae.

Line 337. Fig. 3b – Change the B to lower case.

Response: Text modified. Thanks.

Line 391. Is this statistical analysis some part of the supplementary material? If so, it should be referenced here.

Response: Text modified. We added the supplementary information and modified to "... specifically between 5.45–49.5 mm and 5.6–76.0 mm (Supplementary Fig. 16a and b)". Please see line 364-365.

Line 400. "The distal aspect of the mesonotum in crown cicadids has a cruciform elevation, whereas the rest of cicadoid fossils have scutellum" I don't understand this sentence since cruciform elevation and scutellum are synonyms (Moulds 2005, fig. 3, page 378).

Response: We believe that the cruciform elevation and scutellum are homologous structures, despite exhibiting morphological differences. The scutellum is predominantly an equilateral triangle shape in dorsal view, with (almost) no splitting at the distal end. The cruciform elevation is divided at the distal end, forming a cross-like shape in dorsal view. In the Mesozoic fossil records of Cicadoidea, this structure remains unsplit at the distal end, distinguishing it from the majority of extant Cicadidae species. When using these terms, we will consider their morphological variations and implications variations across time and space. Also because these terms represent homologous structures that we can analyze the macroevolutionary patterns of this structure appearing in fossils and extant taxa.

Line 454. I didn't understand the reference to Supplementary Figure 19 here.

Response: The citation positions of Supplementary Fig. 19 and Supplementary Fig. 18 should be swapped. Text modified to "Cicadidae instead of Tettigarctidae, which has undeveloped tymbal muscles and no resonant abdominal cavity¹⁴ (Supplementary Fig. 19). Since the internal organs, malpighian tubules, and muscles of *Pr. xiai* were preserved (Supplementary Fig. 18)..." Figure 19 showcases light micrographs that compare the abdominal anatomical structures between Tettigarctidae and extant Cicadoidea, offering readers a more intuitive comprehension of the abdominal structures associated with tymbals in extant Cicadoidea.

Line 583. It is usual to mention the full name of the taxon at the first mention in the text.

Response: Text modified. Thanks.

Line 617. It would be interesting to mention the software where the characters were coded.

Response: Thank you for your suggestion. We added "the phylogenetic matrix was coded in software Notepad++". Please see line 781.

Line 623. The root should be stated.

Response: Thank you very much for pointing this out. We added "We chose the type genus *Cycloscytina* of Hylcellidae (Hemiptera) as the root of the tree". Please see line 771.

Phylogeny - I missed a better explanation of the results of the phylogenetic analysis and the characters in the final tree, e.g. which of the characters were recovered as synapomorphies and homoplastic? I didn't find the results relating to the number of recovered trees and an explanation of how the implied weight analysis was performed as well as the calculation of k-values (e.g. SPR similarity) and which k-values were recovered.

Response: Thank you for your questions. We think the characters for the synapomorphies or homoplastic are relative. We discussed certain structures in the discussion section. Please see lines 466-535 "The results combined from Fig. 3a and 3b suggest that *Eunotalia emeryi* gen. et sp. nov., as a stem cicadoid... ". We list the characters and provide phylogenetic reconstruction tree with

characteristics, that is open for further discussion.

The k-values before we mentioned in this Methods section in line 776“...(IW) were used, with concavity values 3 and 12 respectively”. Now we modified the displayed phylogenetic results. We show the trees using equal weighting of characters, and show the unambiguous changes only, slow optimization (DELTRAN) and fast optimization (ACCTRAN) results of the strict consensus tree through the Winclada, which offers a more comprehensive view of the results. Please see in Supplementary Figure 13.

Supplementary Figure 19 – the meaning of the abbreviation “tcav” is missing in the caption.

Response: Text modified to “Abbreviation: tcav, tymbal cavity ...”. Thanks.

Figure 2. I suggest putting on the side of each species’ name the correspondent letter, furthermore the species referent to the letter h is missing in the caption.

Response: Thank you for your suggestion. We have made an attempt to add names beside the letter labels in Fig. 2. However, due to the small font size, they were difficult to discern and somewhat confusing. Therefore, we have decided to maintain the current style of the figure. The legend provides a clear explanation of the specific names. “h” represents *Platypleura kaempferi* (extant Cicadidae), while “c” and “d” represent a male and a female specimen of the species *Pr. xiai*, respectively. The modified text should read as “.Pr. xiai (f and g).”.

Figure 3a. When performing a phylogenetic analysis, it is not usual to use terminals with different ranks. I suggest changing within this figure the terminals represented by genera and subfamilies to species. Moreover, I didn’t find the information of which species were used representing the genera and subfamilies in the analysis, only some of them in the Supplementary Table 2, and in the lines 583 and 600. Besides, the meaning of the numbers in the internal nodes is missing to be informed in the caption. These numbers are not mentioned in the main text (results or discussion).

Response: Thank you very much for your suggestion. We modified and used the terminals in the same ranks for phylogenetic analysis. Please see the phylogenetic results in Supplementary Figure 13. Fig. 3a is a schematic diagram of the phylogenetic relationships. As the phylogenetic placements of the obtained fossils does not fall within the crown cicadoids, and therefore doesn’t involve determining the specific relationships between Mesozoic fossils and genera within Cicadidae, we represent the terminal taxa of the cicadoid crown groups in our study using a broader subfamily-level categorization. We also used the family names of outgroup and the genus names of reported cicadoid fossils in schematic diagram. We feel this might be more intuitive, so still keep the current form.

The numbers simply indicate and represent certain node positions. We have added descriptions of the meanings in the legend, which could potentially lead to further reflection, but this is not related to the results and discussions in the paper. To prevent any misinterpretation of the results, we have temporarily removed these nodes.

Fig. 3b, c, d. I missed a better explanation of what was found as results and what these figures represent.

Response: Referring to your previous suggestion, now we have separately introduced Fig. 3b, c, and d in the Results section. Please see 1.2 Phylogeny and morphological disparity. Figs. 3b and c please line 301-374. Fig. 3d please see the second paragraph of 2.1 Comparisons of extinct and extant species, line 415-422.

REVIEWER #2

The current contribution describes new adult and nymphal cicadoid insects from mid-Cretaceous Kachin amber in Myanmar, achieving a great detail of morphological understanding, presents a phylogeny of the group using both fossil and extant taxa, and provides a series of morphospace analyses using abundant data. The results significantly enrich the knowledge on the evolution of the group, identify tymbal structures and other associated structures such as muscles and a probably resonating cavity for the first time in the fossil record of these insects, allowing to make inferences about their possible sound production ability and communication. The nymphal cicadeoid fossils described —previously only known by an early-instar nymph also from Kachin amber and from a final-instar nymph drawn (yet not described) from slightly younger New Jersey amber (these works cited by the authors in Results), both with fossorial-type forelegs— have allowed the authors to infer and discuss the likely digging and underground habits of these immature insects by comparing with their modern counterparts.

Although the text is generally well-written and makes a compelling case, it contains several language

elements throughout that required improvement. I have tried to mend as many of these as I could, but please make an additional effort to polish this aspect.

Taxa description and analyses are thorough and detailed. Figures are of excellent quality.

Please note that I'm not an hemipterologist nor a native English speaker.

I would like to highlight a few matters that I think are necessary to tackle before the present manuscript can be considered for publication. These, plus the language issues noted above, justify my decision of MINOR-MODERATE reviews for the present work.

Response: Thank you very much for your positive evaluation of this study. We greatly appreciate your detailed modifications and constructive suggestions for improving the article.

1) Need to tone down some claims by adding the proper degree of uncertainty.

First, I would argue that the finding of nymphs with fossorial legs does not represent direct evidence of their habit and behaviour, since these are inferred through morphology and actualism, which my colleagues would agree that they are not devoid of problems. Therefore, I think adding a certain degree of caution and uncertainty is in order, both in abstract (introduction of "had most likely" in line 55) and discussion ("The specialized foreleg of in these fossil nymphs allows to infer their digging behaviour and subterranean lifestyle" rather than "The specialized foreleg of in these fossil nymphs demonstrates their digging behaviour and subterranean lifestyle").

Response: Thank you very much for your suggestion. We agree with your suggestions and have made modifications to the text accordingly. Specific modifications below.

Also, line 525 reads "Cicada nymph fossils with fossorial forelegs here provide one of the few direct fossil records of root-feeding behaviour". Similarly as the case above, I would argue that, since the authors are inferring root-feeding behaviour through the same structures that modern relatives having these fossorial habits possess, the present record does not represent a direct evidence of root-feeding behaviour but an indirect one.

Response: Thank you for pointing this out. The use of the term "direct" does involve a certain level of subjectivity, as it may vary due to differences in reference standards and the degree of definition. We have removed word "direct" and made modifications to this paragraph. Please see lines 657-664 "The evidence for root-feeding is the most poorly documented herbivore functional feeding group in the fossil record63. Fossil evidence among ancient terrestrial arthropods mainly includes borings and fossilized faeces left behind in root fossils from the late Pennsylvanian to Late Triassic (Supplementary Table 12). Cicada nymph fossils with fossorial forelegs here provide one of the few fossil records of root-feeding behaviour. The feeding on roots undoubtedly provides advantages for cicada nymphs to migrate and live many years underground, apparently representing an excellent survival strategy".

Moreover, the fact that all the material comes from a single locality/region from the mid-Cretaceous limits how general and global the conclusions can be, and so this aspect must be treated with caution. The most clear case of this is shown in the sentence around line 470 "species of Cicadoidea were comparatively silent for of* the Mesozoic compared with modern singing cicadas" (see context), which in my opinion has to be changed to "species of Cicadoidea may have been relatively silent during the Mesozoic compared with modern singing cicadas".#.

Response: Thank you very much for pointing out this issue. We agreed that the this aspect should be treated with caution and modified the sentence to "species of Cicadoidea may have been relatively silent during the majority period of Mesozoic compared with modern singing cicadas". However, what we need to emphasize is that while the specimens with well-preserved tymbal structure and three-dimensional bodies in this study predominantly originate from Kachin amber, the fossils spanning from the stem cicadoids to the crown cicadids found within the Kachin amber deposits do not appear to possess developed tymbal-producing systems capable of emitting loud sounds. Considering the morphological disparities of body and forewing fossils from the Jurassic and Triassic, they do not seem to exhibit features that are more closely aligned with crown-group Cicadidae than the fossils from Kachin amber. Hence, we speculate that there might not have been any Cicadoidea fossils from the Jurassic or Triassic with a developed tymbal mechanisms capable of producing loud calls. It seems that for the majority period of the Mesozoic, Cicadoidea were considerably quieter compared to the dominant modern Cicadidae within the cicadoid landscape.

Other instances in which the need to increase the degree of caution/uncertainty in the author's statements is necessary from my point of view are provided in the annotated manuscript.

Response: Thank you for your suggestions and modifications. Based on your comments in the manuscript, we have made further revisions. Our responses to your suggestions and modifications in text have been included at the end of your email comments.

2) Need to revise the format of the systematic section.

I understand that an effort of condensing text and information has been made in order to save space in the 'Systematic palaeontology' section of Results, but in the process, there have been several alterations that affect the understandability of the section and its formal structure in my opinion. "Material" and "Description" subsections should always be directly under a properly introduced species name in which specimens are then listed, not under a genus heading without proper species introduction. The use of "Type genus" and "Type species" is problematic throughout the section. A species should be introduced after the "Type species" has been mentioned under a given genus, so the "Type species" line should not replace the formal introduction of a species (these are not redundant). I would also advise against merging etymology sections of new genera and their only species within (as with *Eunotalia* and *Pranwana*). Lastly, a diagnosis is missing for the genus *Eunotalia* or its only species.

Response: Thank you for your understanding and the recommendation to enhance the standardization of the system description section. Following your advice, we've made adjustments to the format and content of the species description within the main body of the text. "Material" and "Description" subsections were put under the introduced species. We modified the establishment of the type genera to only establish the new genera in this study and added the contents to the section of diagnosis of *Eunotalia* gen. nov. and *E. emeryi* sp. nov.. Please see "Result" section 1.1 Systematic palaeontology.

3. One last matter that I would comment here relates to taphonomy and might impact, to a certain extent, the detailed reconstruction.

Regarding the sentence "Fossils found in Kachin amber and a mass emergence of adult cicadas from the Jurassic Daohugou Lagerstätte demonstrate that they emerged from the soil and ascended to the tree canopy^{33,50}" in line 503, the fossils in Kachin amber could have also been trapped relatively close to the ground level or at the trunk level. The insects would not have necessarily had to reach the tree canopy to get trapped in resin, based on their preservation in amber. Thus, "towards the tree canopy" seems a more suitable expression.

Response: This sentence we want to express is that the discovery of nymphal and various adult fossils in Kachin amber and Daohugou suggests that cicadoid fossils may have occupied diverse ecological niches, with their living environments may range from soil, ground, tree trunks, to tree canopies. We agree with you that insects may not necessarily have reached the tree canopy to be encapsulated, when they were trapped by resin. Here we modified the expression and removed "canopy", please see line 642-646 "Fossils found in Kachin amber and a mass emergence of adult cicadas from the Jurassic Daohugou Lagerstätte demonstrate that they emerged from the soil and ascended upward to an arboreal habitat^{33,61}, implying considerable niche differentiation and a substantial biomass transfer from belowground to above-ground habitats in the mid Mesozoic."

Moreover, roots are known to produce resin in abundance. Are there any taphonomic characteristics of the amber pieces where the cicadoids are preserved that allow to ascertain the aerial origin of the amber pieces and so that the nymphs were actually ascending to or towards the tree canopy? At least some of the amber pieces actually look cloudy due to abundant debris, which is not conclusive of 'litter amber' but might be suggestive of it together with other signs (syninclusions, etc), which might be more indicative of resin entrapment close to the ground. Please consider this option.

Response: Thank you very much for your advice and analysis. Currently, based solely on the preserved fossils in amber, we cannot definitively determine the orientation of the cicada nymphs when they were trapped by the resin, even though some ambers preserve other organisms alongside these cicadoid fossils. We concur with your perspective that nymphs were most likely ensnared by resin near the ground, encompassing scenarios like emerging from the soil, crawling on the ground, ascending tree trunks, or even falling from higher elevations like first instar nymph. Your suggestion offers an interesting research avenue: to observe the type and morphology of accompanying fossils in amber and compare them with inclusions in modern resins. This could help infer the spatial distribution of the amber based on the type and preservation state of the inclusions.

Similarly, for the reconstruction it would be highly advisable to consider the option that fossils could have been trapped by resin fallen to the ground and capturing fauna therein ("litter amber"). Also, it is known that roots can produce resin in abundant quantities (for instance, see Alvarez-Parra et al, 2021, Dinosaur bonebed amber from an original swamp forest soil). Although the taphonomy of root resin production is poorly understood, and there is no evidence --to my knowledge-- yet of insect

entrapment by subaerial resin (that produced by the roots), provided that the current reconstruction showcases the underground layer, not showing this aspect –yet tentatively– would appear to be a missed chance due to the overall detail of the illustration.

Response: Thank you very much for your enlightening suggestion. After considering your suggestions, Mr. Yang Dinghua assisted in adding exuded resin to the surface soil and to a few selected roots in the reconstructed illustration. We agree that some tree roots can secrete resin and form amber, however, it seems to be not very reasonable for cicada nymphs to feed on tree roots that excrete a large amount of resin. Therefore, in this study, the depiction of tree roots secreting resin primarily represents the results of resin secretion caused by fungal infection. Please see Fig. 4.

Specific comments

Line 37. What does represent a “successfully evolved” group of organisms according to the authors? What criteria are used to determine whether a group has evolved successfully or not? Since this might be subjective or perhaps challenging to justify without an explanation, perhaps using an alternative expression is the best and most practical solution.

Response: Thank you very much for your suggestion. We modified this sentence to “Cicadas (Hemiptera: Cicadoidea) are well-known for their evolutionary sound production system and exceptional long-term subterranean habits”. We have selected the most well-known biological behavioral characteristics of cicadas as the content for a more specific introduction. At first, we considered the morphology, behavior, ecology, and biomechanics of cicadas, aspects that have been extensively researched in the realms of culture, life, and science.

Line 39. delete “completely”

Response: Agreed. Text modified.

Line 39. change “taxa” to “groups”

Response: Agreed. Text modified.

Line 40. change “morphological differences that define the modern groups” to “morphological differences defining their extant counterparts”.

Response: Agreed. Text modified.

Line 43. change “make it difficult” to “prevent”

Response: Agreed. Text modified.

Line 61. change “that” to “which”

Response: Agreed. Text modified.

Line 62. delete “evidently”

Response: Agreed. Text modified.

Line 64. change “in slight excess of” to “of slightly more than”

Response: Agreed. Text modified. Please see line 66.

Line 68. delete “designed for”

Response: Agreed. Text modified.

Line 70. delete “of”

Response: Agreed. Text modified.

Line 72. delete “a various”

Response: Agreed. Text modified.

Line 73. delete “were previously unknown”, and add “No previous work has explored’ before ‘the structural...”

Response: Agreed. Text modified.

Line 79. change “as many years” to “more than a decade”

Response: We modified the whole sentence to “Cicada nymphs live underground, tunneling through the substrate and feeding on xylem sap of roots, until the final instar emerges to undergo a final molt, subsequently emerging as a winged adult that achieves sexual maturity before its short-lived

adulthood¹⁵⁻²⁰ and removed this uncertain description.

Line 80. delete “using fossorial behaviours”

Response: Agreed. Text modified.

Line 82. change “a adult with wings” to “a winged adult”

Response: Agreed. Text modified.

Line 84. change “last 13 and 17 years” to “for up to;”

Response: Agreed. We modified the whole sentence to “Cicada nymphs can live underground for up to 17 years^{17, 18}”.

Line 94. change “an inventory that currently is inadequate” to “a record that is clearly insufficient”

Response: Agreed. Text modified.

Line 97. change “the adult fossil records of Cicadoidea include” to “the adult fossil record of Cicadoidea includes”

Response: Agreed. Text modified.

Line 98. change “overly” to “largely”

Response: Agreed. Text modified.

Line 100. change “as Tettigarctidae, based principally on” to “within Tettigarctidae, namely based on”

Response: We changed to “mianly based on...”. Not all fossil records based on forewings, some of them based on hindwing and nymph fossils.

Line 101. delete “that”

Response: Agreed. Text modified.

Line 101. change “In summary,” to “In, sum”

Response: Agreed. Text modified. Please see line 117.

Line 219. modified “CuP fuse a segment with 1A” to “CuP fused with 1A a short distance”

Response: Agreed. Text modified.

Line 248. delete “size of”

Response: Agreed. Text modified.

Line 249. change “...so far; The features including the lateral fissure distinct” to “...so far. The distinct lateral fissure...”

Response: Agreed. Text modified.

Line 251. change “with long common stem of M+CuA make it distinguish from the known genera of Mesozoic cicadoids.” to “with long common stem of M+CuA distinguish the new taxon from other known genera of Mesozoic cicadoids”.

Response: Agreed. Text modified.

Line 254. delete “principally”

Response: Agreed. Text modified.

Line 263. change “Sp.” to “Species”

Response: Agreed. Text modified. Please see line 387.

Line 264. change “2-5” to “2 to 5”

Response: Agreed. Text modified. Please see line 388.

Line 265. The sentence modified to that “These specimens will receive taxonomic treatment elsewhere”

Response: Agreed. Text modified. Please see line 389.

Line 268. change “fossils” to “fossil nymphs”

Response: Agreed. Text modified. Please see line 391.

Line 275. change “Of course, sp. 2 and 3 still possess their own unique features, principally including...” to “Their unique features of species 2 and 3 namely, include..”

Response: Agreed. Text modified. Please see line 399.

Line 288. add “nymphal” before traits

Response: Agreed. Text modified. Please see line 412.

Line 300. change “it is not the first instar nymph” to “it does not represent a first instar nymph”

Response: Agreed. Text modified. Please see line 433.

Line 301. change “has” to “show”

Response: Change “has” to “exhibits”. Please see line 435.

Line 302. change “it is” to “they belonged to”

Response: Agreed. Text modified. Please see line 436.

Line 305. change “apical tooth of the tibia” to “apical tooth on the tibia”

Response: Here “apical tooth of the tibia” here is a formal anatomical term.

Line 306. change “with” to “from”

Response: Agreed. Text modified. Please see line 439.

Line 307. modify “sp.” to “spp.”

Response: Agreed. Text modified. Please see line 440.

Line 318. delete “In this study, we” to “We”

Response: Agreed. Text modified.

Line 322. delete “new genus”

Response: Agreed. Text modified.

Line 336. change “at least by” to “by at least”

Response: Agreed. Text modified. Please see line 466.

Line 337. remove “that”?

Response: Agreed. Text modified.

Line 354. change “For *Pranwanna* gen. nov., it shows more features to crown Cicadidae” to “*Pranwanna* gen. nov. shows more features of crown Cicadidae”

Response: We modified to “The genus *Pranwanna* gen. nov. exhibits a developmental trend in body and forewing morphology closer to the crown cicadids compared to other Mesozoic cicadooids” . Please see line 528.

Line 370. change “discovered” to “found”

Response: Thank you very much for your modification. We adjusted the discussion section and deleted this.

Line 377. delete “at”

Response: Agreed. Text modified.

Line 378. consider replacing “primitive” by “plesiomorphic”

Response: Agreed. Text modified. Please see line 507.

Line 379. I’ m not able to understand the last part of the sentence. Could you please reword?

Response: We modified the text to “These assignments supported the view that extant tettigarctids retains more plesiomorphic characteristics of Mesozoic fossils, such as distinct costal area, separation of CuP and 1A, the length of CuA2 not less than half the length of CuA1, distinct nodal line, RP separation from R earlier than the forking point of M, and the angle of the anal area of the wing lower than the cubital angle.” Please see line 506-510.

Line 383. change to “similar to those of Mesozoic forms than to extant cicadids”.

Response: Agreed. We modified “The wings of Mesozoic fossils are more similar to those of extant tettigarctids than to those of extant cicadids”. Please see line 503.

Line 388. change “we infer ...are..., and the macroevolutionary” to “we posit...were probably., and that the macroevolutionary”. I don’ t think the obtained data allows to ascertain about the rates of evolution of the body and wings in cicadeoids. The authors are free to state their views but with proper degree of caution and uncertainty.

Response: Thank you very much for your modification. We adjusted the discussion section and deleted this. We believe the previous statement was overly broad, and need more detailed, phased analysis and data support. So we remove this sentence and view here.

Line 392. change “that may be common” to “Likely widespread”

Response: Thank you very much for your modification. We adjusted the discussion section and deleted this.

Line 397.delete “are”.

Response: Agreed. Text modified.

Line 407. change “has” to “had”.

Response: Agreed. Text modified. Please see line492.

Line 417. change “ntriguingly, we” to “We”

Response: Agreed. We removed this word.

Line 421. change “... show that the morphological similarities of ...are..” to “...show that the morphology of ...is”

Response: We adjusted the discussion section and modified “Nymphal species 1, 4, and 5 display a morphological similarity that are closer to the crown cicadids than to crown tettigarctids,”. Please see line 543-545.

Lines 428-430. This has been tackled recently in the text and so it feels repetitive here “Similarly, we discovered additional morphological characteristics of cicada nymphs from the Cretaceous and confirm their taxonomic significance”.

Response: Agreed. We remove this sentence.

Line 435. change “In this study, tymbal structures were found in Cicadoidea fossil record for the first time (Fig. 2q–ad), and these structures generally... We discovered tymbal, a chitinous membranous exoskeletal structure (Fig. 2q–x), in the micro-CT scans...” to “We report tymbal structures in fossil Cicadoidea for the first time (Fig. 2q–ad). These paired, chitinous membranous exoskeletal structures are generally.. We discovered tymbal structures (Fig. 2q–x) in the segmented data from micro-CT scans...”

Response: Agreed. Text modified. Please see line 574-576.

Line444. change “is” to “represents”.

Response: Agreed. Text modified. Please see line 582.

Line 445. add “one that” before both males and females.

Response: Agreed. Text modified. Please see line 583.

Line 446. change “have tymbal structures” to “had”.

Response: Agreed. Text modified. Please see line 584.

Line 447. change “discovered” to “found”.

Response: Agreed. Text modified. Please see line 584.

Line 454. add “probably”.

Response: Agreed. Text modified. Please see line 592.

Line 458. add “the” before fossil *Pr. Xiai*

Response: Agreed. Text modified. Please see line 596.

Line 458. add "(Fig. 2aa, ab)" before abdomen.
Response: Agreed. Text modified. Please see line 597.

Line 461 and line 462. change "Fig. 18" to "Fig. 19"
Response: Agreed. Text modified. Please see line 599-600.

Line 465. change "may be" to "may have been".
Response: Agreed. Text modified. Please see line 603.

Line 468. delete "that".
Response: Agreed. Text modified.

Line 471. change to "In any case, species of Cicadoidea may have been relatively silent during the Mesozoic compared with modern singing cicadas".
Response: We modified to "In any case, species of Cicadoidea may have been relatively silent for most period of the Mesozoic compared with modern singing cicadas.". Please see line 608.

Line 474. change "For further confirmation of this inference, it will be ...fossil samples and to test..." to "It will be necessary to...fossil samples and from sources other than Kachin amber to".
Response: Agreed. Text modified. Please see line 610.

Line 478. change "An additionally fascinating discovery is that we found" to "Here we also report".
Response: Agreed. Text modified. Please see line 615.

Line 482. Please here review the previously know fossil record with a sentence, citing references and referring to the Supplementary Material (e.g., Supp Fig 12). Add sentence like "This condition is also present in the final-instar nymphs described from New Jersey amber³⁰".
Response: We modified and added "This feature is also an important identification characteristic for other cicada nymph fossils, from the first instar to the final instar (Supplementary Figure 12)." Please see line 618.

Line 493. change "demonstrates" to "allows to infer".
Response: change to "allows a strong inference regarding". Please see line 630.

Line 497. change "by at least during the" to "by, at least, during".
Response: Agreed. Text modified. Please see line 635.

Line 517. change "reasonable to infer" to "plausible"
Response: Agreed. Text modified. Please see line 654.

Line 530. change "The feeding on roots" to "Root feeding".
Response: Agreed. Text modified. Please see line 668.

Line 532. change "representing an excellent survival strategy" to "representing a successful and highly specialized survival strategy".
Response: Agreed. Text modified. Please see line 670.

Line 552. change "diversify" to "diversified".
Response: Agreed. Text modified. Please see line 690.

Line 556. Change "During this time, cicadoids seem to be undergoing a transition..." to "It is possible that around this time, cicadoids underwent a transition..."
Response: Agreed. Text modified. Please see line 694-695.

Line 561. Change "Morphological measurements indicate that for cicadoids the Mesozoic seems to have a higher morphological space than modern world" to "Our morphological disparity analyses indicate that Mesozoic cicadoids seemed to have occupied a higher morphospace than modern relatives".
Response: Agreed. Text modified. Please see line 702-704.

Line 571. change "veins" to "their venation"

Response: Agreed. Text modified. Please see line 711.

Line 579. change “6 new adults and 5 nymphs” to “six ..and five ...”

Response: Agreed. Text modified. Please see line 721.

Line 582. change “of mid-Cretaceous” to “early Late Cretaceous”

Response: Agreed. Text modified. Please see line 725.

Line 649. change “sd”.

Response: Agreed. Text modified. Please see line 797.

REVIEWER #3

Nature communications

General comments- The authors performed micro-computed tomographic scans on 10 specimens from mid-Cretaceous Kachin amber and extant cicadoids. By combining fossil and extant taxa, and using more detailed anatomical features revealed by micro CT results and morphologically based phylogenetic analysis; forewing and nymphal foreleg, revealed morphological evolutionary tendencies in Cicadoidea. This is a very interesting, original and important approach, however due to the nature of the studied material (fossil), important issues related to the M & M must be clarified. Please check some specific points below.

Geometric morphometrics is a very sensitive technique. Unfortunately, I do not feel confident to evaluate whether the images obtained from specimens in the amber are precise enough for a geometric morphometric analysis. Any possible discrepancies in the position or in the condition of preservation of the analyzed insects could lead to misinterpretations and questionable information as basis for the new genera and species description.

The finding of the first evidence for vocalization associated tymbal structures in fossils, suggesting mid Cretaceous and some other interesting ecological aspects based on the analysis nymphs seems very robust and inedited. However, the scientific bases for the descriptions of new species and genera could be questionable. I presented below some brief suggestions for improving the manuscript but, unfortunately, I don't feel comfortable to either recommend or reject this manuscript for publication.

Response: Thank you for the aspects you've recognized also thank you for the points you've raised with concerns. The description of the new species we've introduced stems from an integration of all existing fossil records, new specimens from this study, and our current understanding of the morphology of contemporary Cicadoidea. Utilizing a multitude of methods, we've endeavored to comprehensively uncover and showcase morphological traits of Cicadoidea fossils to try to enhance our understanding of the phylogenetic relationships, evolutionary history, and ecological significance. Our conclusions are drawn from our present data and insights. We hope this represents a modest step forward, perhaps providing a slightly broader perspective and more fossil data. Please see the detailed answer below.

Specific comments

Abstract- Brief comments on the geographic area where the specimens were found.

Response: Thank you for your suggestion. Text modified to “adult and nymphal fossils from mid-Cretaceous Kachin amber of northern Myanmar”.

Main

Lines 76-77. “In the sentence “live underground for as many years and tunnel through substrate using fossorial behaviors while feeding on xylem sap of roots from the first to fifth instars” and in the Lines 79-80 “ several weeks “. Terms like “many Years and several weeks” should be avoided. The following sentences in the manuscript could be synthesized including the precise information missing in the mentioned lines.”

Response: Thank you very much for pointing this issue. We modified to “Cicada nymphs live underground, tunneling through the substrate and feeding on xylem sap of roots, until the final instar emerges to undergo a final molt, subsequently emerging as a winged adult that achieves sexual maturity before its short-lived adulthood¹⁵⁻²⁰.” Please line 80-83.

Material and Methods

If possible, please provide the geographic coordinates from the sites where the analyzed samples

were found. Please comment whether the references cited for morphological nomenclature were the same ones used for identification and comparisons for the new samples' descriptions.

Response: Thank you very much for your suggestions. We add GPS of the fossil locality in Methods and materials section. Please see line 711 "The geographical coordinates of the site are approximately 26°15'N, 96°34'E.". The terminology used for describing some structures in this study slightly differs from the previous descriptions. Please see "Methods and materials" section, as shown in line 728-735 "The descriptive terminology for the adult structures is mainly based on Evans (1940) and Moulds (2005); and for the nymphal structures, the descriptive terminology is mainly based on Duffels and Ewart (1988) and follow the changed terminology of forewings in this study (Supplementary Note 2 and Supplementary Fig. 7) based on new anatomical observations and a reconciled terminology. The most significant alterations are the discovery of a short RA1 vein of the forewing near the nodal line. The structure name, point of blade of tibia (pbt), is changed to secondary apical tooth of tibia (sapt). Other interpretations please see in Supplementary Note 3.". And please also see Supplementary Figure 7 and Supplementary Note 2. Description for the scheme outlining the forewing venation of Cicadoidea in this study.

I would suggest including in the text, the ten quantifiable morphological characteristics from the body dorsal profile of cicadoid species.

Response: Thank you very much for your suggestion. We selected ten landmarks on the body dorsal profile of cicadoid species, and these ten landmarks might not directly represent ten features. These landmarks, through their relative coordinates, are used to quantify the morphological changes between corresponding body features. Based on your suggestions, we've mentioned some features in the main text, and please line 309-320 "However, these characters have often been overlooked in previous identifications, such as the distance between the compound eyes, the width of the pronotum, the collar width of the pronotum, the position of the anterior lateral angles of the pronotum relative to pronotum width, the exposed width of the mesonotum and the terminal morphology of the mesonotum. We selected ten landmarks to quantify the relative positions of these structures (Supplementary Fig. 14, Table 3). Fig. 3b shows the results of the morphospace for the dorsal features of the head and thorax, obtained through geometric morphometrics..."

It is not clear whether the wing venation of the amber images could be precisely used for a geometric morphometrics analysis. The samples images in the amber are not defined by precision for analysis. Please explore this point for a better description of the methods.

Response: Thank you for posing this question. It's indeed significant as it pertains to the efficacy of applying morphometrics to amber fossil research. Firstly, geometric morphometrics is already used in the study of both amber and rock insect fossils (e.g. Dehon et al., 2019; Shih et al., 2020; Jiang et al., 2023). This involves two issues, but not limited to these two, one is the scale problem of morphological evolution in fossils and extant taxa; another one is, how does the deformation observed in amber fossils compare with the natural morphological variation seen within a species. To answer the second question, we need to understand the extent of morphological variation in the wings of extant specimens of the same species, and the range of variations in a broader taxonomic rank. And to judge the deformation occurs before the entrapment in resin or as a consequence of it. We are conducting more detailed morphometric work on more fossils and extant taxa of Cicadoidea, but this does not affect the work and macroevolutionary patterns (over a long geological time scale) proposed in this paper. Based on our current microscopic observations and micro-CT data, as well as from a taphonomic perspective, the preservation does not appear to affect the overall wing morphology or the identification of key structural features in mid-Cretaceous Kacin amber. We have conducted the disparity analysis of the wing features, including the forewing profiles and venation patterns, of Cicadoidea fossils and extant groups. The overall forewing morphology was analyzed through feature description encoding to discern differences, ensuring the results were not skewed by the deformation. For forewing profiles, we only selected nine landmarks and excluded fossils with significant preservation incompleteness. Even if individual fossils show slight deformation, it doesn't impact our identification of the overall morphological evolutionary trends from fossils to extant groups. Before selecting these features, we first observed all fossil records and a large number of extant groups. Through empirical judgments, we identified some patterns in body and wing morphology changes. By analyzing the selected landmarks or descriptive features, we try to quantify the overall morphology to determine if there are consistent trends of macromorphological shifts.

Dehon, M., Engel, M. S., Gérard, M., Aytekin, A. M., Ghisbain, G., Williams, P. H., Rasmont P. & Michez, D. Morphometric analysis of fossil bumble bees (Hymenoptera, Apidae, Bombini) reveals their taxonomic affinities. *ZooKeys*, 891, 71 (2019).

Shih, P. J., Li, L., Li, D., & Ren, D. Application of geometric morphometric analyses to confirm three new

wasps of Evaniidae (Hymenoptera: Evanioidea) from mid-Cretaceous Myanmar amber. *Cretac. Res.* 109, 104249 (2020).

Jiang, H., Chen, J., & Szweo, J. A new *Jaculistilus* species of Mimarachnidae (Hemiptera: Fulgoromorpha) from mid-Cretaceous Kachin amber of northern Myanmar, with geometric morphometric analysis of the mimarachnid genera. *Cretac. Res.*, 141, 105368 (2023).

Legends- All legends must be self-explanatory. Please provide additional relevant information in the legends for all figs. For instance: geographic location, technique used, In the case of fig 1 geographic location, fig 2 the technique used for the images obtained; fig 3 number of species and specimens illustrated, please name the method used for the phylogenetic analysis etc; Fig 4 please provide basic information on the method used for the image reconstruction.

Response: Thank you very much for your suggestions. Fig. 1 we modified as "...Kachin amber of northern Myanmar". Fig. 2 we modified as "Digital body structures reconstructed from micro-CT data show...". Fig. 3a is the simplified schematic of the phylogenetic relationships, we change the "Schematic diagram of a phylogenetic tree" to "Schematic diagram of a phylogenetic relationships". The specific phylogenetic tree we mentioned see Supplementary Fig. 13 in legend. The methods for phylogenetic analysis were described in the "Methods" section. We also add the basic information of figures preparation and artistic reconstruction in the "Methods" section. Please see line 803-807 "Figures were prepared with the aid of Adobe Photoshop CS6 and CorelDraw 2023. Line drawings were prepared in Procreate. The artistic reconstruction of Figure 4 was created by Mr. Dinghua Yang with aid of Pixologic ZBrush, Maya and Adobe Photoshop CC."

Reviewers' Comments:

Reviewer #1:

Remarks to the Author:

The authors answered and corrected (or justified if not) the issues that I had pointed out in the first version of the manuscript. Some issues regarding the taxonomy section were modified but still need some adjustments. The results and discussion sections are now better presented, but the phylogeny part still needs attention. I strongly recommend the authors to improve these parts of the text before this manuscript can be published.

133. Add "by monotypy" after "sp. nov." in line 146 instead.

145. Add "Description: as for the species". And then add the section "included species" as you did for the other genera. I know you have chosen not to add this for the monotypy genera, but in my opinion, it is important to show this to the reader. A not-taxonomist researcher could read your article and lack understanding what "monotypy" means. That way there is a standard in the text relative to the taxonomy section and you show clearly that the genus has one species listed.

146. Delete "by monotypy".

155. This "Remarks" section is more appropriate for the new genus instead of the new species.

160. "Cretotettigarcta burmensis Fu, Cai and Huang, 2019" or as line 163 Cretotettigarcta bumensis 163 Fu, Cai and Huang, 2019?

168. Revised generic diagnosis from which reference? I suggest informing it between parentheses.

180. Add "Description: as in Fu, Cai, and Huang, 2019".

183. Add a space between the comma and year in "(Jiang et al.,2019)".

189. Same comment as for line 168.

195. Now the taxonomic name is *C. problematica*, correct?

197. Delete "by monotypy". The type of *Cretotettigarcta* is *Cretotettigarcta burmensis* Fu, Cai and Huang, 2019, by monotypy.

215. Before the "Material" section add the section "included species", as commented for line 145.

218. Same comment for line 168. What do you mean with "and also for type species? The diagnosis of a genus is based on characters from all species know for the genus, so it is not necessary to include this comment. However, based on Supplementary Note 1.4. it seems to me you are including new characters in the diagnosis and description based on a new specimen. If so, I suggest including a comment in the Remarks section of the genus, for example "We include new characters based on a new specimen in the diagnosis of the genus ..."

227. This description is relative to a new specimen found of *Vetuprosbole parallelica*, so it is not a description of the genus. I suggest to change for "Description: as for the species (adapted from Fu, Cai and Huang, 2019)", and add a new paragraph (section) for *Vetuprosbole parallelica* with a revised diagnosis and/or description, including new information found in this new specimen, including the "Remarks" section you wrote for the genus.

235. Again, include in this line the section "included species". After diagnosis add "Description: as for the species".

252. "by monotypy" should be in the line 233. Delete it in the line 252.

263. "The distinct...?"

267. 1.2 Phylogeny and morphological disparity. You based the description of some results of the analyses in the diagram. I don't understand what the proposal of this diagram is, as you wrote "to better illustrate the relationships between the fossils and extant taxa", but we can see the relationships in the consensus tree. You have relationships recovered in the trees of Supplementary Figure 13 that in the diagram seem to be in polytomy. For example *Shuraboprosbole*, *Sanmai*, and *Macrotettigarcta*. I strongly suggest that you replace the diagram by the consensus tree in figure 3a. Figure 3a. I disagree that using family and subfamilie names in the diagram is intuitive. For me it is confusing, because I don't know what species in Supplementary Figure 13 corresponds to the family and subfamily terminals in the diagram. The same way in the matrix. The best choice will be to standardize the taxa as in the Supplementary Figure 13 allowing comparisons. Using the family, subfamily, or genus as terminal taxa in the figures or in the matrix means all taxa within the rank

were examined and analyzed. This is not the case and therefore the figure contains wrong information. I suggest, again, to modify it.

Supplementary Figure 13. I liked how you present this figure now. My only suggestion is indicating the family name in each terminal or clade.

269. "The data matrix was primarily analyzed using the maximum parsimony method, with the results being validated through Bayesian inference". This sentence needs to be rephrased. Maximum parsimony and Bayesian inference are different methods, the first is the optimality criteria behind the cladistic method, and the second is a probabilistic method. So, the term "validated" is not appropriate, because we have phylogenetics hypothesis based on two different methods.

285. "The results from maximum parsimony and Bayesian inference are relatively consistent (Supplementary Fig. 13)." What do you mean with "consistent"? Similar? I do not think "consistent" is the best term to use here. If you mean that the results are similar, I suggest "congruent".

Please, inform the quantity of trees recovered using the maximum parsimony. Based on supplementary figure 13 I believe there are three trees recovered, but it should be clear in the text.

286. "In both analyses, Eunotalia gen. nov. was placed at the base of the cicadoid stem". Here we have three points to verify. Firstly, "placed" seems to me you put the species there and it is not the appropriate term. The terminals are recovered by the analyses. Secondly, it is important to state the name of the species (*Eunotalia ameryigen*) not the genus names. You did not analyze the genus, but the species. See my comment for the figure 3a. Please, modify it in the rest of the text. Thirdly, in my opinion "base" is not the appropriate term, because the trees are not fixed, you can twist it and the position of top or base are irrelevant. I suggest using, per example, "*Eunotalia ameryigen*" was recovered closest to the outgroup".

287. "Tianyuprosbole was placed closest to the clade of extant tettigarctids and can be grouped together into a clade with the latter." This sentence is confusing. *Tianyuprosbole zhengi* was recovered as the sister-group of the extant tettigarctids forming a clade. You can mention the synapomorphies shared by these species.

289. "Shuraboprosbole, Sanmai, and *Macrotettigarcta* from Daohugou, as well as *Cretotettigarcta*, *Vetuprosbole*, and *Pranwanna* gen. nov. from Kachin amber can be grouped into a large clade with extant cicadids." The term "can be grouped" is not appropriate. You performed two phylogenetic analysis and you have the evolutionary hypothesis, so the terminals were recovered in branches forming clades. When you described the results of a phylogenetic analysis you need to describe the relationships recovered between the terminals. So, the species were recovered as...but not "can be grouped". They were here recovered in determined positions in the tree, and future studies involving phylogenetic analysis with other terminals or other methods can recover one or more different evolutionary hypothesis.

294. "The groups from Kachin amber were placed closer to the extant cicadids." You described it in a different way in the line 290.

389. "These specimens will receive taxonomic treatment elsewhere". Treatments should be in plural, no? I don't understand where these treatments will be.

452. I don't understand this sentence. You are mixing the phylogenetic relationships discussion with the NMDS. Besides, these terms "before" and "previously" are not commonly used in phylogenetic relationships discussion. My suggestion is to rewrite the phylogenetic discussion.

1152. "Complete phylogenetic trees see Supplementary Fig. 13." Add "For" at the beginning.

Reviewer #2:

Remarks to the Author:

I thank the authors for having considered my comments and adjusted the manuscript accordingly.

I would still advise that the manuscript's English, particularly the new text, is carefully checked for improvement and mending any existing mistakes.

Reviewer #3:
Remarks to the Author:
NCOMMS-23-22805A

The reviewed version of the manuscript was greatly improved.
All comments and questions were well and detailed answered.
The manuscript can now be recommended for publication.

Reviewer #1 (Remarks to the Author):

The authors answered and corrected (or justified if not) the issues that I had pointed out in the first version of the manuscript. Some issues regarding the taxonomy section were modified but still need some adjustments. The results and discussion sections are now better presented, but the phylogeny part still needs attention. I strongly recommend the authors to improve these parts of the text before this manuscript can be published.

Response: Thank you very much for the further review and suggestions. We appreciate your feedback and have made improvements accordingly. Please see the details below.

133. Add “by monotypy” after “sp. nov.” in line 146 instead.

Response: Agreed. Text modified. Please see line 133.

145. Add “Description: as for the species”. And then add the section “included species” as you did for the other genera. I know you have chosen not to add this for the monotypy genera, but in my opinion, it is important to show this to the reader. A not-taxonomist researcher could read your article and lack understanding what “monotypy” means. That way there is a standard in the text relative to the taxonomy section and you show clearly that the genus has one species listed.

Response: Thank you very much for your suggestion and explanation. We agree and have added “Description: as for the species”.

146. Delete “by monotypy”.

Response: Agreed. Text modified.

155. This “Remarks” section is more appropriate for the new genus instead of the new species.

Response: Thank you for pointing this out. We agree and have moved this section to the part about the new genus.

160. “Cretotettigarcta burmensis Fu, Cai and Huang, 2019” or as line 163 Cretotettigarcta bumensis 163 Fu, Cai and Huang, 2019?

Response: Thank you for pointing this out. We have deleted “by monotypy”.

168. Revised generic diagnosis from which reference? I suggest informing it between parentheses.

Response: Agreed. Text modified to “Diagnosis (revised from Fu et al., 2019)”. Please see line 170.

180. Add “Description: as in Fu, Cai, and Huang, 2019”.

Response: Agreed. Text modified.

183. Add a space between the comma and year in “(Jiang et al.,2019)”.

Response: Text modified. Thanks.

189. Same comment as for line 168.

Response: Text modified. Thanks.

195. Now the taxonomic name is *C. problematica*, correct?

Response: Yes. Now the taxonomic name is *C. problematica*.

197. Delete “by monotypy”. The type of *Cretotettigarcta* is *Cretotettigarcta burmensis* Fu, Cai and Huang, 2019, by monotypy.

Response: Agreed. Text modified. Thanks.

215. Before the “Material” section add the section “included species”, as commented for line 145.

Response: Agreed. Text modified. Thanks.

218. Same comment for line 168. What do you mean with “and also for type species? The diagnosis of a genus is based on characters from all species known for the genus, so it is not necessary to include this comment. However, based on Supplementary Note 1.4. it seems to me you are including new characters in the diagnosis and description based on a new specimen. If so, I suggest including a comment in the Remarks section of the genus, for example “We include new characters based on a new specimen in the diagnosis of the genus ...”.

227. This description is relative to a new specimen found of *Vetuprosbole parallelica*, so it is not a description of the genus. I suggest to change for “Description: as for the species (adapted from Fu, Cai and Huang, 2019)”, and add a new paragraph (section) for *Vetuprosbole parallelica* with a revised diagnosis and/or description, including new information found in this new specimen, including the “Remarks” section you wrote for the genus.

Response: Thank you very much for your explanations and suggestions. We modified “Revised generic diagnosis (and also for type species)” to “Diagnosis (adapted from Fu et al., 2019)”. In the Remarks, we have mentioned “new features found in the new material of *V. parallelica*, including...”. Please see line 232. We have added “Included species: *Vetuprosbole parallelica* Fu, Cai, and Huang, 2019.” before Material section and added “Description: as for the species of the new specimen”. Please see lines 218 and 231.

235. Again, include in this line the section “included species”. After diagnosis add “Description: as for the species”.

Response: Agreed. Text modified.

252. “by monotypy” should be in the line 233. Delete it in the line 252.

Response: Agreed. Text modified.

263. “The distinct...”?

Response: Text modified. Thanks.

267. 1.2 Phylogeny and morphological disparity. You based the description of some results of the analyses in the diagram. I don’t understand what the proposal of this diagram is, as you wrote “to better illustrate the relationships between the fossils and extant taxa”, but we can see the relationships in the consensus tree. You have relationships recovered in the trees of Supplementary Figure 13 that in the diagram seem to be in polytomy. For example *Shuraboprosbole*, *Sanmai*, and *Macrotettigarcta*. I strongly suggest that you replace the diagram by the consensus tree in figure 3a.

Response: Thank you very much for pointing this out. Figure 3a illustrates our main point that some Mesozoic cicadoid fossils previously attributed to "Tettigarctidae" including specimens from the Middle Jurassic and mid-Cretaceous, may represent stem lineages transitioning towards modern Cicadidae. This suggests that the previously considered "Tettigarctidae" which included Mesozoic fossils, is actually a paraphyletic group. The schematic diagram includes a timeline displaying the geological time in which these fossils are preserved, important for discussing the evolution of these taxa. Including those concerning *Shuraboprobole*, *Sanmai*, and *Macrotettigarcta*, different potential relationships have been recovered among these three fossils based on the phylogenetic results. However, the specific relationships among these fossils lack clear support. Nevertheless, this does not affect or change the main conclusions and discussions presented above. We have chosen to retain multiple possible outcomes in the supplementary information and refrain from depicting any specific relationships that were less supportive in the main text figures. Thank you very much for your suggestion again. We have modified Figure 3a and kept terminal taxa at the species level.

Figure 3a. I disagree that using family and subfamilie names in the diagram is intuitive. For me it is confusing, because I don't know what species in Supplementary Figure 13 corresponds to the family and subfamily terminals in the diagram. The same way in the matrix. The best choice will be to standardize the taxa as in the Supplementary Figure 13 allowing comparisons. Using the family, subfamily, or genus as terminal taxa in the figures or in the matrix means all taxa within the rank were examined and analyzed. This is not the case and therefore the figure contains wrong information. I suggest, again, to modify it.

Response: Thank you for reemphasizing this point and thank you very much for your detailed explanations. We agree and have modified figure 3a and keep terminal taxa at the species level.

Supplementary Figure 13. I liked how you present this figure now. My only suggestion is indicating the family name in each terminal or clade.

Response: Thank you foracknowledging our modifications. We have added the subfamily and/or family names to some terminals. The new specimens and Mesozoic cicadoiods used here have not yet been assigned to a specific family level yet by our work and are currently categorized under the superfamily Cicadoidea. This study primarily aims to elucidate the relationships between Mesozoic fossils and modern taxa. The assignment of family-level classifications needs be confirmed in more precise taxonomic work and careful consideration. We also welcome further discussion and research from interested scholars regarding the classification of these taxa.

269. "The data matrix was primarily analyzed using the maximum parsimony method, with the results being validated through Bayesian inference". This sentence needs to be rephrased. Maximum parsimony and Bayesian inference are different methods, the first is the optimality criteria behind the cladistic method, and the second is a probabilistic method. So, the term "validated" is not appropriate, because we have phylogenetics hypothesis based on two different methods.

Response: Thank you very much for pointing this out and explanations. We have modified it to "the data matrix was analysed using the maximum parsimony method and Bayesian inference".

285. "The results from maximum parsimony and Bayesian inference are relatively consistent (Supplementary Fig. 13)." What do you mean with "consistent"? Similar? I do not think "consistent" is

the best term to use here. If you mean that the results are similar, I suggest “congruent”.

Response: Thank you very much for pointing this out. We agree and have modified “consistent” to “congruent”.

Please, inform the quantity of trees recovered using the maximum parsimony. Based on supplementary figure 13 I believe there are three trees recovered, but it should be clear in the text.

Response: We mentioned the information in Phylogeny section of Methods. “Four trees were obtained and the strict consensus tree was used with equal weighting parsimony⁹⁴...Supplementary Fig. 13 displayed the unambiguous changes only, slow optimization and fast optimization results of the strict consensus tree⁹⁶”. Please see line 788.

286. “In both analyses, *Eunotalia* gen. nov. was placed at the base of the cicadoid stem”. Here we have three points to verify. Firstly, “placed” seems to me you put the species there and it is not the appropriate term. The terminals are recovered by the analyses. Secondly, it is important to state the name of the species (*Eunotalia ameryigen*) not the genus names. You did not analyze the genus, but the species. See my comment for the figure 3a. Please, modify it in the rest of the text. Thirdly, in my opinion “base” is not the appropriate term, because the trees are not fixed, you can twist it and the position of top or base are irrelevant. I suggest using, per example, “*Eunotalia emeryi*” was recovered closest to the outgroup”.

Response: Thank you very much for pointing this out. The expressions used here indeed had issues. We have modified “placed” to “recovered” in the text, modified the sentence to “*Eunotalia emeryi* sp. nov. was recovered at the stem group of the cicadoids”.

287. “*Tianyuprosbole* was placed closest to the clade of extant tettigarctids and can be grouped together into a clade with the latter.” This sentence is confusing. *Tianyuprosbole zhengi* was recovered as the sister-group of the extant tettigarctids forming a clade. You can mention the synapomorphies shared by these species.

Response: Thank you very much for pointing this out and suggestions. We have modified to “*Tianyuprosbole zhengi* was recovered as the sister-group of the extant tettigarctids.”

289. “*Shuraboprosbole*, *Sanmai*, and *Macrotettigarcta* from Daohugou, as well as *Cretotettigarcta*, *Vetuprosbole*, and *Pranwanna* gen. nov. from Kachin amber can be grouped into a large clade with extant cicadids.” The term “can be grouped” is not appropriate. You performed two phylogenetic analysis and you have the evolutionary hypothesis, so the terminals were recovered in branches forming clades. When you described the results of a phylogenetic analysis you need to describe the relationships recovered between the terminals. So, the species were recovered as...but not “can be grouped”. They were here recovered in determined positions in the tree, and future studies involving phylogenetic analysis with other terminals or other methods can recover one or more different evolutionary hypothesis.

Response: Thank you very much for pointing this out and for explanations. We agree and have modified the “can be grouped” to “can be recovered”.

294. “The groups from Kachin amber were placed closer to the extant cicadids.” You described it in a different way in the line 290.

Response: Thank you very much for pointing this out. We have modified to “ The groups, *Cretotettigarcta*, *Vetuprosbole*, and *Pranwanna* gen. nov., were recovered closer to the extant cicadids. ”

389. “These specimens will receive taxonomic treatment elsewhere”. Treatments should be in plural, no? I don’t understand where these treatments will be.

Response: Thank you very much for pointing this out. Upon further consideration, we have removed this sentence. Our morphological analysis can reveal trends in the classification of cicada nymph fossils. However, due to the limited number of fossils and the quality of morphological preservation, it is challenging to establish relatively precise classification positions that correspond to adult fossils. In this study, we have presented as much morphological information as possible from nymph fossils. Further classification results may require the discovery of more adult and nymph fossils, as well as the analysis of extant taxa to comprehensively confirm their classification positions. We believe that the work of determining more detailed classification positions requires further research.

452. I don’t understand this sentence. You are mixing the phylogenetic relationships discussion with the NMDS. Besides, these terms “before” and “previously” are not commonly used in phylogenetic relationships discussion. My suggestion is to rewrite the phylogenetic discussion.

Response: Thank you very much for pointing this out. We have rewritten the phylogenetic discussion, and please see from line 450. In the discussion, we need to compare the relationship between fossils and extant groups as previously thought with the results and views in this study. Therefore, we used “before” and “previously”. We mention Fig. 3c because we want to show and compare the phylogenetic relationships and the classification of groups based on forewing features. Before this study, fossils of the Mesozoic cicadoids were primarily classified based on forewing structures. We have adjusted the discussion of Fig. 3c, please see from line 496.

1152. “Complete phylogenetic trees see Supplementary Fig. 13.” Add “For” at the beginning.

Response: Agreed. Text modified.

Reviewer #2 (Remarks to the Author):

I thank the authors for having considered my comments and adjusted the manuscript accordingly.

I would still advise that the manuscript's English, particularly the new text, is carefully checked for improvement and mending any existing mistakes.

Response: Thank you very much for your feedback and advice again. We have further improved the manuscript.

Reviewer #3 (Remarks to the Author):

NCOMMS-23-22805A

The reviewed version of the manuscript was greatly improved.

All comments and questions were well and detailed answered.
The manuscript can now be recommended for publication.

Response: Thank you very much again for your assessment of the revised manuscript.